# TransferTraj: A Vehicle Trajectory Learning Model for Region and Task Transferability

**Tonglong Wei**[1],[*] **Yan Lin**[2],[*] **Zeyu Zhou**[1], **Haomin Wen**[3], **Jilin Hu**[4]
**Shengnan Guo**[1,5],[†] **Youfang Lin**[1,6], **Gao Cong**[7], **Huaiyu Wan**[1,6]
[1]School of Computer Science and Technology, Beijing Jiaotong University, China
[2]Department of Computer Science, Aalborg University, Denmark
[3]Carnegie Mellon University
[4]School of Data Science and Engineering, East China Normal University, China
[5] Key Laboratory of Big Data & Artificial Intelligence in Transportation, Ministry of Education, China
[6] Beijing Key Laboratory of Traffic Data Mining and Embodied Intelligence, China
[7]College of Computing and Data Science, Nanyang Technological University
{weitonglong, zeyuzhou, guoshn, yflin, hywan}@bjtu.edu.cn, lyan@cs.aau.dk,
haominwe@andrew.cmu.edu, jlhu@dase.ecnu.edu.cn, gaocong@ntu.edu.sg

## Abstract

Vehicle GPS trajectories provide valuable movement information that supports various downstream tasks and applications. A desirable trajectory learning model should be able to transfer across regions and tasks without retraining, avoiding the need to maintain multiple specialized models and subpar performance with limited training data. However, each region has its unique spatial features and contexts, which are reflected in vehicle movement patterns and are difficult to generalize. Additionally, transferring across different tasks faces technical challenges due to the varying input-output structures required for each task. Existing efforts towards transferability primarily involve learning embedding vectors for trajectories, which perform poorly in region transfer and require retraining of prediction modules for task transfer.

To address these challenges, we propose *TransferTraj*, a vehicle GPS trajectory learning model that excels in both region and task transferability. For region transferability, we introduce RTTE as the main learnable module within TransferTraj. It integrates spatial, temporal, POI, and road network modalities of trajectories to effectively manage variations in spatial context distribution across regions. It also introduces a TRIE module for incorporating relative information of spatial features and a spatial context MoE module for handling movement patterns in diverse contexts. For task transferability, we propose a task-transferable input-output scheme that unifies the input-output structure of different tasks into the masking and recovery of modalities and trajectory points. This approach allows TransferTraj to be pre-trained once and transferred to different tasks without retraining. We conduct extensive experiments on three real-world vehicle trajectory datasets under various transfer settings, including task transfer, zero-shot region transfer, and few-shot region transfer. Experimental results demonstrate that TransferTraj significantly outperforms state-of-the-art baselines in different scenarios, validating its effectiveness in region and task transfer. Code is available at https://github.com/wtl52656/TransferTraj.

---

[*]Both authors contributed equally to this research.
[†]Corresponding author.

# 1 Introduction

A vehicle GPS trajectory is a sequence of (location, time) pairs that record a vehicle's movement. The widespread adoption of location-aware devices—such as in-vehicle navigation systems and smartphones—has led to the large-scale collection of vehicle trajectory data. This growing availability, together with increasing interest in intelligent transportation systems (ITS), encouraging the development of trajectory learning models that can perform various generative tasks to power real-world applications in ITS, such as trajectory prediction [38, 42, 39], trajectory recovery [26, 5, 36, 34], travel time estimation [37, 43, 24, 28], and trajectory generation [44, 45, 35]. These tasks typically involve inputting trajectories with missing features or points and generating these missing components.

A common approach for addressing various trajectory generative tasks is to train a dedicated trajectory learning model for each region and task. However, this practice leads to substantial training costs and requires that each region have a large amount of available trajectory data to support model training. To mitigate these limitations, it is desirable to develop a trajectory learning model with both region and task transferability. This means it should be able to transfer across regions and tasks without retraining. For example, region transferability allows a model trained in region A to work effectively in region B, while task transferability enables a model developed for trajectory prediction to also perform well for travel time estimation. This transferability has two key benefits. First, it eliminates the need to retrain and maintain multiple dedicated models, improving overall computational efficiency. Second, transferring a trained model to other regions or tasks with limited training data can be more effective than training from scratch. However, existing efforts have struggled to develop a trajectory learning model with effective region and task transferability due to the following challenges.

**The challenge of region transferability** arises from differences in spatial features and the distribution of spatial context (including POIs and roads) across regions, leading to significant variations in vehicle movement patterns, such as turning patterns, accelerating behaviour, and movement directions. Specifically, each region has its own geographical size, which makes common spatial features standardization methods, such as normalization [41, 31] and grid discretization [18, 39], prone to producing inconsistent spatial scales across regions. Therefore, a model trained on the spatial features of one region cannot be directly applied to another. Moreover, trajectories with the same travel purpose across regions—such as trips from parks to residential areas via main roads, indicating a shared purpose of returning home after leisure—their movement patterns often vary substantially due to the distribution difference of spatial context. These discrepancies prevent the direct transfer of movement patterns learned from one region to another.

**The challenge of task transferability** stems from the differences in input-output structures and the correlations learned for different tasks, making it difficult for a model to handle various types of tasks simultaneously. For instance, the trajectory prediction task involves generating future trajectory points based on historical trajectory sequences, focusing on learning sequential correlations between trajectory points. In contrast, the travel time estimation task predicts arrival time from origin to destination, emphasizing temporal correlations between origin-destination pairs and travel times. These tasks exhibit substantial discrepancies in their input-output structure and learned correlations. Existing efforts towards task transferability mostly adhere to the embedding strategy, which learns trajectory encoders for mapping vehicle trajectories into embedding vectors [11, 21, 15, 22]. Although these embedding vectors contain movement information of trajectories, they still necessitate prediction modules for adaptation to downstream tasks, which require additional training and storage of parameters.

In this paper, we propose **TransferTraj**, a trajectory learning model that can be pretrained on one region and effectively transferred to other regions and various types of generative tasks. TransferTraj encompasses two core components: a Region-Transferable Trajectory Encoder (*RTTE*) and a task-transferable input-output scheme. RTTE enables TransferTraj region transferability. It incorporates spatial, temporal, POI, and road network modalities of trajectories to discern differences in spatial context distribution across regions. Additionally, it includes a Trajectory Relative Information Extraction (*TRIE*) module for transferable modeling of relative relation in spatial features across different regions and a Spatial Context Mixture-of-Experts (*SC-MoE*) module to identify and share movement patterns under similar spatial contexts. The task-transferable input-output scheme equips TransferTraj with task transferability by unifying the input-output structure of different generative tasks into the masking and recovery of modalities and trajectory points. Coupled with a pre-training mechanism, this approach enables TransferTraj to adapt to various tasks without retraining. We

evaluate the effectiveness of TransferTraj through extensive experiments on three real-world vehicle GPS trajectory datasets with diverse settings. In terms of task transferability, only with pre-training, TransferTraj outperforms the SOTA baselines by 7.94% to 20.18% across different tasks. And it achieves an average improvement of 83.70% and 33.68% in zero-shot region transferability.

## 2 Related Works

We categorize vehicle trajectory learning models into non-transferable and transferable, based on their ability to generalize across different regions or tasks.

**Non-transferable trajectory learning models** are typically trained in an end-to-end fashion for specific tasks and regions. For instance, trajectory prediction methods such as DeepMove [10], HST-LSTM [17], and ACN [25] leverage Recurrent Neural Networks (RNN) [13, 7] to capture sequential correlations in trajectories, while PreCLN [39] employs Transformers [30] for processing vehicle trajectories. In the realm of trajectory recovery, methods like TrImpute [8] and DHTR [33] capture spatiotemporal relationships of sparse trajectories to infer missing GPS coordinates, while MTrajRec [26], RNTrajRec [5], and MM-STGED [36] simultaneously recover missing points and map them to road networks. For origin-destination travel time estimation, approaches such as TEMP [32], MURAT [19], DeepOD [43], and DOT [24] estimate travel time using origin, destination, and departure time information. Although these methods are straightforward to implement, their limited transferability necessitates separate model designs and training for different tasks and regions, significantly increasing computational and storage demands.

**Transferable trajectory learning models** primarily employ pre-trained embedding techniques to enable task transferability. Models such as trajectory2vec [41], t2vec [18], Trembr [11], START [15], and MMTEC [22] develop trajectory encoders that map vehicle trajectories into embedding vectors using pre-training techniques like auto-encoding [12] and contrastive learning [3]. Despite their versatility, these trajectory encoders still require additional prediction modules to generate task-specific predictions from the embedding vectors. Furthermore, they remain non-transferable across regions due to inherent differences in spatial features and context distribution.

## 3 Preliminaries

**Vehicle trajectory.** A vehicle trajectory $\mathcal{T}$ is a sequence of trajectory points: $\mathcal{T} = \langle p_1, p_2, \ldots, p_n \rangle$, where $n$ is the number of points. Each point $p_i = (\lng_i, \lat_i, t_i)$ consists of the longitude $\lng_i$, latitude $\lat_i$, and timestamp $t_i$, representing the vehicle's location at a specific time.

**POI.** A point of interest (POI) is a significant geographical location with specific cultural, environmental, or economic importance. We represent a POI as $l = (\lng, \lat, \desc^l)$, where $\lng$ and $\lat$ denote the coordinates of the POI, and $\desc^l$ is a textual description including the name, type, and address of the POI.

**Road segment.** We represent a road segment as $r = (\lng, \lat, \desc^r)$, where $\lng$ and $\lat$ denote the coordinates of the midpoint of the road segment, and $\desc^r$ is a textual description including the name, type, and length of the road segment.

**Problem definition.** *Region and task transferable vehicle trajectory learning* aims to develop a trajectory learning model $f_\theta$ with learnable parameters $\theta$. Once pre-trained, this model should effectively transfer between trajectory datasets from different regions and accurately generate the required outputs for various tasks based on their inputs, without needing to retrain the parameters $\theta$.

## 4 Methodology

In this paper, we introduce TransferTraj, a trajectory learning model that excels in region and task transferability. The framework of TransferTraj is shown in Figure 1. It comprises two key components: the *Region-Transferable Trajectory Encoder* (**RTTE**) and *task-transferable input-output scheme*.

RTTE is the learnable module of TransferTraj, designed to achieve region transferability. It first maps each trajectory point into four distinct feature modalities: spatial, temporal, point of interest (POI), and road network. The inclusion of POI and road network modalities enables the model to capture

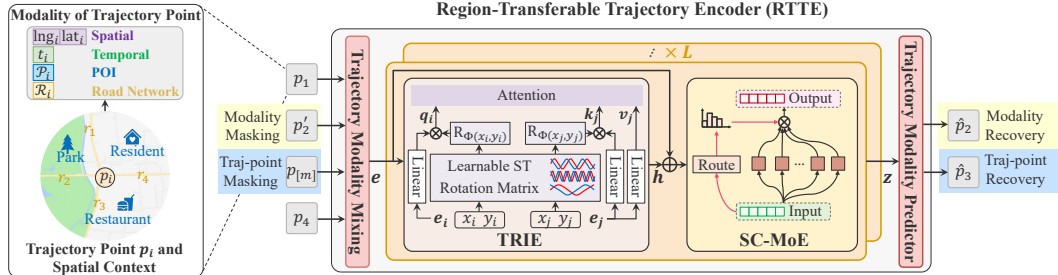

Figure 1: The framework of TransferTraj.

differences in spatial context distributions across regions. A trajectory modality mixing layer then integrates these four modalities into a latent vector representing the trajectory point. Next, the latent vectors are fed into $L$ stacked layers to extract trajectory features. Each layer contains a *Trajectory Relative Information Extraction* (**TRIE**) block and a *Spatial Context Mixture-of-Experts* (**SC-MoE**) block. The TRIE block, inspired by RoFormer [29], captures relative spatial relationships among trajectory points, preventing the model from becoming biased towards specific regions. The SC-MoE block dynamically selects appropriate movement pattern experts based on the local spatial context, allowing the model to adapt flexibly to diverse contexts.

The task-transferable input-output scheme equips TransferTraj with the flexibility to handle various tasks. It standardizes the input and output structure of different tasks by combining two approaches: masking and recovering modalities or trajectory points. In one approach, spatial or temporal modalities of a trajectory point are masked in the input and then recovered in the output. In another approach, entire trajectory points are masked with special tokens in the input and subsequently recovered in the output. This scheme enables TransferTraj to transfer seamlessly between tasks without re-training after the pre-training phase.

## 4.1 Region-Transferable Trajectory Encoder (RTTE)

We begin by representing each point $p_i$ in a trajectory $\mathcal{T}$ as a tuple of four modalities: spatial, temporal, POI, and road network, denoted as $p_i = ((\lng_i, \lat_i), t_i, \mathcal{P}_i, \mathcal{R}_i)$. The POI modality $\mathcal{P}_i = \{l \mid \dis(l, p_i) \leq \varphi_{\text{dist}}^{\text{poi}}\}$ is defined as the set of all POIs lying within a distance $\varphi_{\text{dist}}^{\text{poi}}$ of the trajectory point $p_i$. Similarly, the road network modality $\mathcal{R}_i = \{r \mid \dis(r, p_i) \leq \varphi_{\text{dist}}^{\text{road}}\}$ is the set of all road segments located within a distance $\varphi_{\text{dist}}^{\text{road}}$ of $p_i$.

### 4.1.1 Trajectory Modality Mixing

**For the spatial modality**, we compute each point's position relative to the first point in the trajectory, i.e., $(x_i, y_i) = (\lng_i - \lng_1, \lat_i - \lat_1)$. A linear layer is then applied to obtain the spatial modality embedding $\mathbf{e}_i^s \in \mathbb{R}^d$. This approach preserves the scale variations of different regions better than methods like min-max normalization or grid discretization. **For the temporal modality**, each timestamp $t_i$ is represented by a 4-dimensional vector: day of the week, hour of the day, minute of the hour, and the time difference in minutes relative to $t_1$. These features are encoded using a learnable Fourier encoding layer [20] and then projected into a $d$-dimensional embedding $\mathbf{e}_i^t$. **For the POI and road network modalities**, we utilize a pre-trained text embedding model[3] to encode the textual descriptions of POIs and road segments. The resulting embeddings are processed via mean pooling and linear transformations to obtain fixed-dimensional vectors, denoted as $\mathbf{e}_i^p \in \mathbb{R}^d$ and $\mathbf{e}_i^r \in \mathbb{R}^d$. This approach enables the model to capture region-independent semantic information, enhancing its transferability across different geographic areas.

Next, the latent vectors of the four modalities at each trajectory point are organized into a sequence of length 4 and fed into the Transformer. This is followed by a mean pooling layer that merges information across modalities to compute the embedding vector $\mathbf{e}_i \in \mathbb{R}^d$ for trajectory point $p_i$.

---

[3]https://platform.openai.com/docs/guides/embeddings

Following this, the mixed embedding vectors $e_i$ are passed through $L$ stacked identical layers to capture correlations among trajectory points. Each layer comprises a TRIE block and a SC-MoE block. The output of the $(l-1)$-th layer serves as the input to the $l$-th layer. The first layer receives the sequence of embedding vectors for all trajectory points, formulated as $\langle e_1, \cdots, e_n \rangle$, where $n$ is the number of points. We use the first layer as an example to describe the TRIE and SC-MoE blocks in detail.

### 4.1.2 Trajectory Relative Information Extraction (TRIE)

To capture the spatiotemporal dependencies among trajectory points while enabling transferability across regions, we propose modeling the correlations between trajectory points' embedding vectors and relative spatial information simultaneously. Extracting this relative information prevents the model from becoming biased towards specific regions, thereby providing a region-transferable approach for handling spatial features.

Specifically, for a trajectory point with spatial modality $(x, y)$, we first generate its spatial representation vector $\Phi(x, y) = W_\Phi(x \| y)$ through a learnable linear transformation, where $\|$ denotes vector concatenation and $W_\Phi \in \mathbb{R}^{d/2 \times 2}$ is a learnable projection matrix. We then introduce a learnable spatio-temporal rotation matrix to encode spatial information, calculated as follows:

$$\mathbf{R}_{\Phi(x,y)} = \begin{bmatrix} \cos\phi_1(x,y)\theta_1 & -\sin\phi_1(x,y)\theta_1 & \cdots & 0 & 0 \\ \sin\phi_1(x,y)\theta_1 & \cos\phi_1(x,y)\theta_1 & \cdots & 0 & 0 \\ \vdots & \vdots & \ddots & \vdots & \vdots \\ 0 & 0 & \cdots & \cos\phi_{d/2}(x,y)\theta_{d/2} & -\sin\phi_{d/2}(x,y)\theta_{d/2} \\ 0 & 0 & \cdots & \sin\phi_{d/2}(x,y)\theta_{d/2} & \cos\phi_{d/2}(x,y)\theta_{d/2} \end{bmatrix} \in \mathbb{R}^{d \times d} \quad (1)$$

where $\phi_k(x, y)$ is the $k$-th element of $\Phi(x, y)$, $\theta_1, \theta_2, \ldots, \theta_{d/2}$ are frequency weights, and $\theta_k = 10000^{-2k/d}$.

Next, we apply the rotation matrix to query $q_i$ and key $k_j$ in the attention mechanism to model the relationships between embeddings $e_i$ and $e_j$. Formally,

$$q_i = \mathbf{R}_{\Phi(x_i,y_i)} W_q e_i, \qquad k_j = \mathbf{R}_{\Phi(x_j,y_j)} W_k e_j, \qquad v_j = W_v e_j, \quad (2)$$

where $W_q \in \mathbb{R}^{d \times d}$, $W_k \in \mathbb{R}^{d \times d}$, and $W_v \in \mathbb{R}^{d \times d}$ are mapping matrices. Following this, we implement the attention mechanism for the input sequence and calculate the $i$-th hidden state as $h_i$.

Through the above process, our TRIE addresses regional transferability with two key advantages: (1) TRIE enhances the model's understanding of relative spatial information by naturally considering the true relative distances $\Phi(x_i - x_j, y_i - y_j)$ between trajectory points during attention computation through a rotational mechanism. A detailed proof of this enhancement is provided in Appendix C. (2) By leveraging the continuous and periodic nature of trigonometric functions in the learnable spatiotemporal rotation matrix, TRIE can generalize to trajectories longer than those seen during training and capture their relative information effectively.

### 4.1.3 Spatial Context Mixture-of-Experts (SC-MoE)

To effectively capture the movement patterns of trajectories, we observe that such patterns are strongly influenced by the spatial context. For instance, highways and regions with sparse Points of Interest (POIs) typically correspond to high-speed, straight-line movement, whereas dense POI areas with complex road networks often result in stop-and-go behavior. While the global distribution of spatial context differs between regions, there exist similar local spatial contexts. Motivated by this observation, we introduce a Spatial Context Mixture of Experts (SC-MoE) that integrates multiple experts to model diverse movement patterns under different spatial contexts, while learning similar movement patterns in similar contexts. Specifically, given the learned trajectory point embedding $h_i$ in the TRIE layer and the modality embedding $e_i$, the output of SC-MoE is:

$$h_i' = \sum_{j=1}^C G_j(h_i + e_i) E_j(h_i + e_i), \quad (3)$$

where $G(\cdot)$ denotes the gating network and $E_j(\cdot)$ denotes the output of the $j$-th expert network, each expert network is implemented by two MLP layers. There are a total of $C$ expert networks, each with separate parameters, and the gating network outputs a $C$-dimensional vector.

To prevent the movement patterns from being represented by the same set of experts and to explore better expert combinations in varying contexts, we implement noisy top-K gating [27] before applying

the softmax function in the gating network. This mechanism routes each spatial context to the most suitable experts, guided by the gating network, thereby distinguishing movement patterns across different contexts. The noisy top-K gating for the trajectory point $p_i$ is formulated as:

$$
\begin{aligned}
G(\boldsymbol{h}_i + \boldsymbol{e}_i) &= \text{Softmax}(\text{TopK}(H(\boldsymbol{h}_i + \boldsymbol{e}_i), k)), \\
H(\boldsymbol{h}_i + \boldsymbol{e}_i)_j &= ((\boldsymbol{h}_i + \boldsymbol{e}_i) \cdot W_g)_j + \mathcal{N}(0,1) \cdot \text{Softplus}(((\boldsymbol{h}_i + \boldsymbol{e}_i) \cdot W_{noise})_j), \\
\text{TopK}(\boldsymbol{a}, k)_j &= \begin{cases} \boldsymbol{a}_j & \text{if } a_j \text{ is in the top } k \text{ elements of } \boldsymbol{a} \\ -\infty & \text{otherwise.} \end{cases}
\end{aligned}
\tag{4}
$$

After stacking $L$ TRIE and SC-MoE layers, we obtain the latent vector of the trajectory point $p_i$, denoted as $\boldsymbol{z}_i$. We then feed $\boldsymbol{z}_i$ into the trajectory modality predictor to predict the trajectory point's spatial and temporal modalities.

### 4.1.4 Trajectory Modality Predictor

We use a linear projection layer to predict the $i$-th trajectory point's spatial modality $(\hat{x}_i, \hat{y}_i)$ and obtain the coordinates $(\hat{\text{lng}}_i, \hat{\text{lat}}_i)$ by adding the coordinates of the first point. For the temporal modality, we use a linear projection layer followed by Softplus activation to predict the temporal features $\hat{\boldsymbol{t}}_i$. To supervise the predicted modalities, we apply the Mean Squared Error (MSE) loss function to the predicted spatial and temporal modalities, formulated as follows:

$$
\mathcal{L}_i^s = (\hat{x}_i - x_i)^2 + (\hat{y}_i - y_i)^2, \qquad \mathcal{L}_i^t = \|\hat{\boldsymbol{t}}_i - \boldsymbol{t}_i\|_2
\tag{5}
$$

### 4.2 Task-Transferable Input-Output Scheme

To enable task transferability, we standardize the input-output structure of various generative tasks, allowing seamless adaptation across different tasks. Our approach combines two methods: 1) masking a specific modality of a single trajectory point in the input and recovering it in the output, and 2) masking an entire trajectory point from the input using a special mask point and subsequently recovering it in the output.

Given a trajectory point $p_i = ((\text{lng}_i, \text{lat}_i), t_i, \mathcal{P}_i, \mathcal{R}_i)$ with its four modalities, we can mask either the spatial or temporal modality. A point with a masked spatial modality is represented as $p_i^{ms} = ([m], t_i, [m], [m])$, where the spatial, POI, and road network modalities are replaced by the special mask token $[m]$. Note that since the POI and road network modalities are linked to the spatial modality, masking the spatial modality necessarily masks these associated modalities as well. Similarly, a point with a masked temporal modality is denoted as $p_i^{mt} = ((\text{lng}_i, \text{lat}_i), [m], \mathcal{P}_i, \mathcal{R}_i)$. The fully masked trajectory point, where all modalities are masked, is represented as $p_{[m]} = ([m], [m], [m], [m])$. The masked modalities in $p_i^{ms}$, $p_i^{mt}$, or $p_{[m]}$ are then recovered in the corresponding output of the trajectory modality predictor module at the same time step, denoted as $\hat{p}_i$.

**Pre-training with the scheme.** We propose to pre-train TransferTraj with a mixture of the above two masking and recovery approaches, enabling it to transfer effectively between tasks without requiring re-training. Given a trajectory $\mathcal{T}$ in the pre-training dataset, we first randomly select the starting point $s$ and ending point $e$ for sub-trajectory masking from a uniform distribution $U(1, n)$, where $n$ is the trajectory length and $s < e$. For each trajectory point in the selected sub-trajectory, $\langle p_s, p_{s+1}, \ldots, p_e \rangle$, complete trajectory point masking is applied in the input, and these points must be fully recovered in the output. For the remaining trajectory points in the input, $p_1, \ldots, p_{s-1}, p_{e+1}, \ldots, p_n$, we randomly mask either their spatial or temporal modalities with an equal probability, and task the model with recovering these masked modalities in the output. Finally, the pre-training loss for trajectory $\mathcal{T}$ is computed as the sum of reconstruction losses across all masked and recovered modalities.

By integrating our modality and trajectory point masking and recovery scheme with the pre-training procedure, TransferTraj can adapt to various tasks without retraining.

## 5 Experiments

To evaluate the performance of our proposed model, we conduct extensive experiments on three real-world vehicle trajectory datasets across three generative tasks: Trajectory Prediction (TP), Trajectory Recovery (TR), and Origin-Destination Travel Time Estimation (OD TTE).

The **TP** task aims to forecast the future segment of a trajectory given its historical segment. For this task, the historical portion of the trajectory can be provided to TransferTraj as $\langle p_1, p_2, \ldots, p_{n'}, \{p_{[m]}\}^{n-n'} \rangle$, where $n'$ is the historical length, and $\{p_{[m]}\}^{n-n'}$ indicates that there are $n - n'$ points represented by $p_{[m]}$. The future portion is then predicted. We set $n' = n - 5$, and evaluate the precision of the trajectories' destinations. MAE and RMSE of the shortest distance on the Earth's surface are used as evaluation metrics.

The **TR** task aims to reconstruct the dense trajectory with a finer sampling interval, given the sparse trajectory. For this task, we define the sampling interval for the dense trajectory as $\epsilon$ and for the sparse trajectory as $\mu$, where $\mu > \epsilon$. Therefore, a sequence $\langle p_1, \{p_{[m]}\}^{\frac{\mu}{\epsilon}}, p_{1+\frac{\mu}{\epsilon}}, \{p_{[m]}\}^{\frac{\mu}{\epsilon}}, p_{1+\frac{\mu}{\epsilon}*2}, \cdots, p_n, \rangle$ can be used as input to TransferTraj, and the missed trajectory points are derived from the recovered spatial modality of the masked trajectory points. We set $\mu = 4 \cdot \epsilon, 8 \cdot \epsilon$, and $16 \cdot \epsilon$, respectively. MAE and RMSE are used as evaluation metrics.

The **OD TTE** task aims to predict the travel time of a trajectory given its starting and ending locations and departure time. For this task, a sequence $\langle p_1, p'_n \rangle$ can be used as the input to TransferTraj, where $p'_n = ((\lng_n, \lat_n), [m], \mathcal{P}_n, \mathcal{R}_n)$. The predicted travel time is derived from the recovered temporal modality of the last point $\hat{p}_n$. MAE, RMSE, and MAPE are used as evaluation metrics.

**Datasets.** In our experiments, we use three real-world vehicle trajectory datasets derived from Chengdu, Xi'an, and Porto. Since the original trajectories in the Chengdu and Xi'an datasets have very dense sampling intervals, we employ a three-hop resampling method to retain only a portion of the trajectory points, ensuring that most trajectories have sampling intervals of at least 6 seconds. We also filter out trajectories containing fewer than 5 or more than 120 trajectory points. Additionally, we retrieve information on POIs and road networks within these datasets' areas of interest from the AMap API[4] and OpenStreetMap. Table 8 presents the statistics of these datasets after preprocessing.

**Baselines.** For the TP task, we compare with the latest trajectory representation learning methods, including t2vec [9], Trembr [11], CTLE [23], Toast [4], TrajCL [1], START [15], and LightPath [40]. For the TR task, we compare with Linear, MPR [6], TrImpute [8], DHTR [33], MTrajRec [26], RNTrajRec [5], and MM-STGED [36]. For the OD TTE task, we compare with RNE [14], TEMP [32], LR, GBM, ST-NN [16], MuRAT [19], DeepOD [43], and DOT [24]. For our TransferTraj, we introduce two variants for comparison: (1) TransferTraj (wo pt), which excludes the pre-training process and trains the model using only task-specific methods; and (2) TransferTraj (wo ft), which omits the fine-tuning stage and directly handles specific tasks in a zero-shot manner after pre-training.

**Setting.** TransferTraj is first pre-trained for 30 epochs on the training set using the trajectory masking and recovery strategy to enhance task transferability. We then perform task-specific fine-tuning to further improve performance on downstream tasks. The model is trained on one dataset and then transferred to the other two datasets. Additional implementation details are provided in Appendix B.2.

Table 1: Overall performance of methods on trajectory prediction.

| Dataset | Chengdu | | Xian | | Porto | |
|---|---|---|---|---|---|---|
| Metric / Method | RMSE ↓ (meters) | MAE ↓ (meters) | RMSE ↓ (meters) | MAE ↓ (meters) | RMSE ↓ (meters) | MAE ↓ (meters) |
| Trembr (wo ft) | 1787.18 ± 29.01 | 1419.58 ± 28.95 | 2067.80 ± 16.30 | 1749.76 ± 18.82 | 1640.13 ± 14.88 | 1270.43 ± 16.13 |
| START (wo ft) | 1347.13 ± 30.72 | 1111.77 ± 29.11 | 1406.06 ± 18.42 | 1173.62 ± 17.18 | 1258.09 ± 19.78 | 1038.36 ± 19.57 |
| LightPath (wo ft) | 2365.87 ± 57.52 | 1948.97 ± 57.78 | 2177.27 ± 60.03 | 1859.35 ± 48.50 | 2345.71 ± 55.28 | 2036.93 ± 43.78 |
| t2vec | 579.30 ± 11.94 | 387.50 ± 4.03 | 482.64 ± 2.67 | 310.08 ± 3.00 | 360.90 ± 0.83 | 212.92 ± 2.44 |
| Trembr | 505.62 ± 4.57 | 376.88 ± 7.34 | 473.97 ± 1.24 | 301.45 ± 4.98 | 315.50 ± 4.90 | 182.38 ± 1.43 |
| CTLE | 430.19 ± 52.64 | 382.82 ± 52.88 | 477.70 ± 48.25 | 384.08 ± 53.18 | 319.85 ± 59.83 | 179.93 ± 36.71 |
| Toast | 480.52 ± 82.39 | 412.58 ± 72.32 | 523.76 ± 67.04 | 443.99 ± 60.41 | 482.58 ± 66.46 | 290.55 ± 74.33 |
| TrajCL | 365.50 ± 19.14 | 272.63 ± 25.32 | 383.39 ± 7.30 | 262.20 ± 10.68 | 327.10 ± 1.48 | 176.47 ± 1.50 |
| START | 333.10 ± 10.47 | 240.40 ± 15.10 | 319.00 ± 4.27 | 208.35 ± 7.30 | 260.29 ± 3.68 | 159.73 ± 29.09 |
| LightPath | 553.27 ± 42.26 | 360.86 ± 56.41 | 598.20 ± 15.57 | 348.61 ± 19.32 | 388.46 ± 22.46 | 217.04 ± 38.44 |
| TransferTraj (wo pt) | 289.25 ± 9.19 | 218.43 ± 5.63 | 296.32 ± 5.71 | 197.79 ± 5.08 | 249.33 ± 7.41 | 164.10 ± 6.72 |
| TransferTraj (wo ft) | 223.15 ± 9.28 | 176.88 ± 9.45 | 238.17 ± 4.34 | 174.59 ± 7.36 | 218.49 ± 4.77 | 153.26 ± 5.27 |
| **TransferTraj** | 187.91 ± 8.65 | 144.53 ± 5.25 | 212.62 ± 2.79 | 154.86 ± 3.94 | 196.46 ± 4.81 | 149.75 ± 3.46 |

Red denotes the best result, and blue denotes the second-best result. ↓ means lower is better.

---

[4] https://lbs.amap.com/api/javascript-api-v2

Table 2: Overall performance of methods on trajectory recovery on the Chengdu dataset. Performance on Xi'an and Porto as shown in Table 9 and 10.

| Sampling Intervals | $\mu = \epsilon * 4$ | | $\mu = \epsilon * 8$ | | $\mu = \epsilon * 16$ | |
|---|---|---|---|---|---|---|
| Metric / Method | RMSE ↓ (meters) | MAE ↓ (meters) | RMSE ↓ (meters) | MAE ↓ (meters) | RMSE ↓ (meters) | MAE ↓ (meters) |
| Linear | 280.39 | 188.04 | 347.92 | 247.20 | 503.96 | 398.67 |
| MPR | 266.06 | 173.55 | 332.68 | 233.17 | 494.28 | 375.19 |
| TrImpute | 242.36 | 164.96 | 316.08 | 216.83 | 476.83 | 350.19 |
| DHTR | 273.76 ± 4.77 | 176.09 ± 3.42 | 328.26 ± 4.74 | 223.81 ± 4.47 | 487.54 ± 5.22 | 357.22 ± 4.38 |
| MTrajRec | 214.07 ± 5.19 | 143.16 ± 4.06 | 301.81 ± 6.02 | 208.13 ± 7.10 | 438.48 ± 5.43 | 335.90 ± 8.04 |
| RNTrajRec | 205.29 ± 3.92 | 138.47 ± 2.48 | 262.93 ± 3.74 | 193.06 ± 4.19 | 420.04 ± 5.86 | 316.70 ± 8.73 |
| MM-STGED | 181.03 ± 8.27 | 130.24 ± 9.58 | 247.46 ± 6.30 | 186.07 ± 5.12 | 405.29 ± 5.51 | 301.81 ± 8.22 |
| TransferTraj (wo pt) | 194.48 ± 3.50 | 139.40 ± 4.69 | 228.46 ± 8.12 | 174.20 ± 6.37 | 361.05 ± 4.43 | 286.20 ± 4.63 |
| TransferTraj (wo ft) | 158.39 ± 5.89 | 115.23 ± 5.26 | 194.69 ± 6.76 | 149.51 ± 6.41 | 321.06 ± 7.88 | 237.10 ± 6.38 |
| **TransferTraj** | 135.15 ± 2.35 | 97.87 ± 3.17 | 176.91 ± 2.57 | 129.26 ± 4.91 | 277.32 ± 2.83 | 200.38 ± 3.90 |

Red denotes the best result, and blue denotes the second-best result. ↓ means lower is better.

Table 3: Overall performance of methods on OD TTE.

| Dataset | Chengdu | | | Xian | | | Porto | | |
|---|---|---|---|---|---|---|---|---|---|
| Metric / Method | RMSE ↓ (minutes) | MAE ↓ (minutes) | MAPE ↓ (%) | RMSE ↓ (minutes) | MAE ↓ (minutes) | MAPE ↓ (%) | RMSE ↓ (minutes) | MAE ↓ (minutes) | MAPE ↓ (%) |
| RNE | 3.663±0.367 | 3.478±0.406 | 20.236±2.917 | 6.651±1.104 | 5.355±0.990 | 16.817±1.519 | 4.466±0.783 | 2.712±0.219 | 34.261±5.927 |
| TEMP | 3.493±0.929 | 3.016±1.399 | 17.602±2.119 | 6.731±1.328 | 5.403±1.207 | 17.288±2.229 | 4.214±0.720 | 2.693±0.829 | 34.520±6.294 |
| LR | 3.478±0.216 | 2.939±0.287 | 15.386±2.369 | 6.169±1.035 | 5.002±0.723 | 16.295±0.793 | 4.039±1.114 | 2.516±0.781 | 31.676±4.130 |
| GBM | 3.412±0.338 | 2.703±0.297 | 14.927±0.925 | 5.538±0.022 | 4.720±0.036 | 15.720±0.917 | 3.991±0.319 | 2.397±0.211 | 27.396±1.172 |
| ST-NN | 3.379±0.628 | 2.685±0.419 | 14.429±2.914 | 5.362±0.330 | 4.619±0.414 | 15.319±1.190 | 3.739±0.610 | 2.284±0.593 | 24.761±2.047 |
| MuRAT | 3.330±0.110 | 2.621±0.125 | 13.198±2.280 | 5.028±1.530 | 4.332±1.294 | 13.885±3.245 | 3.691±0.235 | 2.029±0.459 | 20.619±3.197 |
| DeepOD | 3.219±0.038 | 2.579±0.046 | 12.730±2.592 | 4.715±0.720 | 3.818±0.620 | 12.028±2.199 | 3.028±0.056 | 1.936±0.043 | 17.398±0.992 |
| DOT | 3.161±0.402 | 2.390±0.105 | 10.193±0.936 | 4.394±0.946 | 3.296±0.729 | 9.504±0.366 | 2.782±0.031 | 1.794±0.014 | 16.306±2.104 |
| TransferTraj(wo pt) | 3.063±0.133 | 2.486±0.188 | 9.984±1.316 | 4.005±0.041 | 2.795±0.143 | 9.082±1.038 | 2.649±0.721 | 1.593±0.761 | 15.631±2.406 |
| TransferTraj(wo ft) | 2.992±0.383 | 2.265±0.010 | 9.613±0.690 | 4.061±0.290 | 3.003±0.205 | 9.219±2.483 | 2.580±0.400 | 1.321±0.008 | 15.967±3.429 |
| **TransferTraj** | 2.861±0.171 | 2.060±0.112 | 9.360±0.529 | 3.816±0.428 | 2.569±0.236 | 8.343±0.997 | 2.138±0.011 | 1.225±0.006 | 14.682±0.829 |

Red denotes the best result, and blue denotes the second-best result. ↓ means lower is better.

## 5.1 Performance Comparison

**Task transferability.** Tables 1, 2, and 3 present the performance of various methods on the TP, TR, and OD TTE tasks. Our proposed model consistently outperforms all baselines across tasks. Using pre-training alone (TransferTraj wo ft), our model achieves average **20.18%**, **17.87%**, and **7.94%** gains over the SOTA baselines START, MM-STGED, and DOT on three tasks, demonstrating strong task transferability. When further fine-tuned on the target tasks, performance further improves by **11.41%**, **13.59%**, and **8.62%**, respectively. In the TP task, we also freeze the pre-trained encoders of trajectory representation methods and fine-tune only the prediction heads (wo ft). The results show a significant performance degradation compared to the fully fine-tuned approach. To reach optimal performance, these methods require fine-tuning the entire trajectory encoder with task supervision, thus failing to fully achieve task transferability. By comparison, TransferTraj does not require fine-tuning the learnable model or prediction modules to reach the reported performance. It can be pre-trained once and directly perform different tasks with high performance. This demonstrates its superior task transferability, enabling high efficiency and practical utilization in real-world applications.

Table 4: Zero-shot region transfer performance of methods on trajectory prediction.

| Dataset | Chengdu → Xian | | Chengdu → Porto | | Xian → Porto | | Xian → Chengdu | | Porto → Chengdu | | Porto → Xian | |
|---|---|---|---|---|---|---|---|---|---|---|---|---|
| Metric / Method | RMSE ↓ (meters) | MAE ↓ (meters) | RMSE ↓ (meters) | MAE ↓ (meters) | RMSE ↓ (meters) | MAE ↓ (meters) | RMSE ↓ (meters) | MAE ↓ (meters) | RMSE ↓ (meters) | MAE ↓ (meters) | RMSE ↓ (meters) | MAE ↓ (meters) |
| t2vec (wo ft) | 3258.54 | 2994.46 | 2963.71 | 2200.58 | 2890.09 | 2177.13 | 2920.99 | 2560.44 | 2679.04 | 2263.09 | 2900.75 | 2257.04 |
| Trembr (wo ft) | 2914.69 | 2566.69 | 2445.43 | 1881.04 | 2677.14 | 1860.29 | 2706.51 | 2234.04 | 2598.04 | 2093.15 | 2675.06 | 2142.97 |
| CTLE (wo ft) | 3775.45 | 3479.44 | 3260.91 | 2741.78 | 3173.62 | 2267.60 | 3131.03 | 2753.10 | 2956.23 | 2590.10 | 3285.04 | 2602.12 |
| Toast (wo ft) | 2622.43 | 2100.58 | 2200.25 | 1662.52 | 2262.42 | 1811.06 | 2673.71 | 2294.38 | 2237.22 | 1875.89 | 2502.75 | 1926.15 |
| TrajCL (wo ft) | 2144.53 | 1823.11 | 1888.65 | 1130.30 | 1797.26 | 1580.88 | 2278.60 | 1718.00 | 1582.46 | 1297.13 | 1681.19 | 1274.26 |
| START (wo ft) | 1891.36 | 1627.67 | 1963.76 | 1241.50 | 1655.80 | 1369.25 | 1859.58 | 1569.15 | 1669.19 | 1214.12 | 1768.68 | 1376.46 |
| LightPath (wo ft) | 2793.72 | 2465.34 | 2025.05 | 1315.76 | 1881.52 | 1525.59 | 2610.30 | 2228.35 | 1811.92 | 1460.01 | 1946.95 | 1555.08 |
| **TransferTraj** | 329.77 | 242.83 | 294.38 | 216.19 | 286.24 | 209.99 | 225.19 | 170.64 | 239.60 | 188.21 | 357.45 | 294.46 |

Red denotes the best result, and blue denotes the second-best result. ↓ means lower is better.

Table 5: Zero-shot region transfer performance of methods on OD TTE.

| Dataset | Chengdu → Xian | | | Chengdu → Porto | | | Xian → Porto | | |
|---|---|---|---|---|---|---|---|---|---|
| Metric / Method | RMSE ↓ (minutes) | MAE ↓ (minutes) | MAPE ↓ (%) | RMSE ↓ (minutes) | MAE ↓ (minutes) | MAPE ↓ (%) | RMSE ↓ (minutes) | MAE ↓ (minutes) | MAPE ↓ (%) |
| LR | 7.982 | 6.912 | 21.793 | 5.280 | 3.118 | 38.191 | 4.991 | 3.094 | 37.286 |
| GBM | 7.317 | 6.106 | 19.435 | 4.720 | 3.002 | 32.957 | 4.804 | 2.946 | 34.342 |
| ST-NN | 6.322 | 5.661 | 17.036 | 4.394 | 2.819 | 28.911 | 4.522 | 2.800 | 29.160 |
| MuRAT | 5.917 | 4.613 | 15.307 | 3.905 | 2.694 | 27.395 | 4.099 | 2.705 | 24.038 |
| DeepOD | 5.076 | 4.003 | 13.428 | 3.706 | 2.585 | 22.021 | 3.954 | 2.639 | 21.488 |
| DOT | 4.957 | 3.560 | 12.743 | 3.334 | 2.317 | 18.881 | 3.214 | 2.527 | 19.768 |
| **TransferTraj** | 4.484 | 3.236 | 10.532 | 2.762 | 1.987 | 17.428 | 2.675 | 2.002 | 18.284 |

| Dataset | Xian → Chengdu | | | Porto → Chengdu | | | Porto → Xian | | |
|---|---|---|---|---|---|---|---|---|---|
| Metric / Method | RMSE ↓ (minutes) | MAE ↓ (minutes) | MAPE ↓ (%) | RMSE ↓ (minutes) | MAE ↓ (minutes) | MAPE ↓ (%) | RMSE ↓ (minutes) | MAE ↓ (minutes) | MAPE ↓ (%) |
| LR | 4.923 | 3.901 | 23.977 | 5.384 | 4.321 | 21.118 | 7.186 | 6.510 | 20.469 |
| GBM | 4.890 | 3.884 | 20.027 | 5.002 | 4.103 | 20.915 | 6.952 | 5.914 | 18.325 |
| ST-NN | 4.774 | 3.723 | 19.903 | 4.948 | 3.965 | 18.736 | 5.796 | 5.104 | 16.259 |
| MuRAT | 4.525 | 3.527 | 16.043 | 4.677 | 3.833 | 17.118 | 5.413 | 4.803 | 15.405 |
| DeepOD | 4.497 | 3.474 | 14.338 | 4.532 | 3.621 | 16.001 | 5.305 | 4.172 | 15.927 |
| DOT | 4.206 | 3.362 | 12.621 | 4.334 | 3.356 | 13.699 | 4.995 | 4.203 | 13.892 |
| **TransferTraj** | 3.842 | 2.973 | 11.320 | 3.995 | 3.168 | 11.935 | 4.874 | 3.961 | 12.396 |

Red denotes the best result, and blue denotes the second-best result. ↓ means lower is better.

Table 6: Ablation Study on Chengdu dataset.

| Task | TP | | TR | | OD TTE | | |
|---|---|---|---|---|---|---|---|
| Metric / Method | RMSE ↓ (meters) | MAE ↓ (meters) | RMSE ↓ (meters) | MAE ↓ (meters) | RMSE ↓ (minutes) | MAE ↓ (minutes) | MAPE ↓ (minutes) |
| wo TRIE | 236.18 | 185.14 | 163.47 | 124.66 | 3.347 | 2.861 | 11.685 |
| wo SC-MoE | 227.36 | 167.75 | 166.48 | 127.45 | 3.240 | 2.585 | 11.584 |
| wo POI modality | 226.25 | 159.37 | 152.19 | 119.38 | 3.166 | 2.307 | 11.388 |
| wo road network modality | 223.59 | 165.44 | 164.07 | 124.53 | 3.128 | 2.221 | 11.631 |
| **TransferTraj** | 197.91 | 144.53 | 135.15 | 97.87 | 2.861 | 2.060 | 9.360 |

Red denotes the best result, and blue denotes the second-best result. ↓ means lower is better.

**Region transferability.** We conduct regional transfer experiments under both zero-shot and few-shot settings. In the zero-shot setting, TP and OD TTE baselines are pre-trained on one region and directly applied to other regions. For the TR task, we do not conduct zero-shot transfer experiments, as the baselines rely on multiclassification over road IDs, and the number of roads varies across regions, making direct transfer infeasible. In the few-shot setting, all models are first pre-trained on one region and then fine-tuned on other regions by randomly sampling 5,000 trajectories.

We present the performance comparison of zero-shot transfer in Tables 4 and 5 and few-shot transfer in Tables 11, 12, 13, 14 and 15. The experimental results demonstrate that our model consistently outperforms baseline methods across all three tasks, highlighting its strong region transferability. Specifically, on the TP task, our method achieves improvements of **83.70%** and **33.68%** over SOTA models in zero-shot and few-shot settings. For the TR task, it improves performance by **18.08%** in the few-shot setting, and for the OD TTE task, the gains are **10.88%** and **13.07%** in zero-shot and few-shot settings. We attribute this success to the model's ability to capture the relative spatial information of trajectories and to extract shared movement patterns from the surrounding context.

## 5.2 Model Analysis

**Ablation study.** The experimental results presented in Table 6 reveal several important findings. First, removing the TRIE component leads to a significant performance decline, highlighting the critical importance of capturing spatial relative information between trajectory points for effective trajectory understanding. Furthermore, our ablation study on SC-MoE demonstrates a substantial performance drop of **14.52%** on the OD TTE task, validating its effectiveness in capturing complex movement patterns. Additionally, the integration of POI and road network modalities further enhances the model's performance by providing a richer spatial context for trajectory analysis.

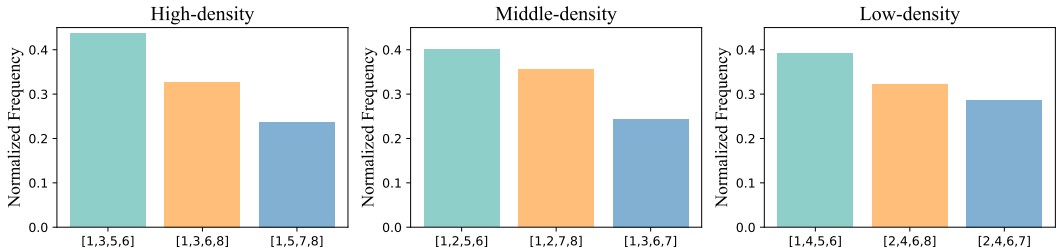

Figure 2: Expect activation distribution.

**Analysis of the SC-MoE.** We analyze the experts activated across different spatial contexts by grouping trajectory points based on the density of nearby POIs and road segments. Specifically, we classify the local spatial context into three categories: high-density regions ($> 15$ POIs and road segments), medium-density regions ($5 \sim 15$), and low-density regions ($< 5$). For each category, we compute the distribution of activated experts, as shown in Figure 2. The results reveal distinct expert activation distributions across density levels, indicating that the model dynamically selects specialized combinations of experts to effectively capture movement patterns in diverse spatial contexts.

Table 7: Efficiency of methods on TP task.

| Dataset | Chengdu | | | Xi'an | | | Porto | | |
|---|---|---|---|---|---|---|---|---|---|
| t2vec | 1.64 | 2.78 | 4.45 | 6.30 | 5.94 | 9.70 | 7.21 | 9.29 | 13.52 |
| Trembr | 5.75 | 3.36 | 3.23 | 5.30 | 6.07 | 9.72 | 6.14 | 10.68 | 13.03 |
| CTLE | 3.76 | 4.53 | 14.58 | 3.76 | 14.35 | 33.86 | 3.76 | 18.55 | 53.13 |
| Toast | 4.01 | 4.40 | 14.54 | 3.56 | 10.65 | 33.86 | 3.95 | 17.64 | 52.88 |
| TrajCL | 4.38 | 7.70 | 10.25 | 3.93 | 14.57 | 23.88 | 4.46 | 19.50 | 39.15 |
| START | 15.93 | 15.93 | 28.70 | 15.03 | 37.53 | 49.89 | 17.81 | 62.88 | 67.49 |
| LightPath | 12.96 | 10.25 | 22.49 | 12.51 | 23.22 | 46.26 | 13.90 | 46.49 | 63.65 |
| **TransferTraj** | 3.64 | 1.27 | 1.21 | 3.64 | 4.09 | 3.92 | 3.83 | 8.38 | 8.17 |

Red denotes the best result, and blue denotes the second-best result.

**Efficiency Study** Table 7 compares the efficiency of different methods on three datasets in TP task, measured by the size of the learning model and the time required for training and testing. We observe that TransferTraj is lightweight, with a model size comparable to RNN-based methods like t2vec and Trembr, and much smaller than state-of-the-art methods like START and LightPath. TransferTraj also has an efficient training process and is competitive at test time compared to other methods. Overall, TransferTraj enhances the efficiency of real-world applications, as it only needs to be trained once and can perform various tasks in different regions without re-training.

# 6 Conclusion

We propose TransferTraj, a vehicle trajectory learning model that excels in both region and task transferability. First, we introduce RTTE as TransferTraj's learnable component to enable region transferability. This component incorporates POI and road network modalities, enabling the model to comprehend spatial context distributions across diverse regions. Equipped with TRIE and SC-MoE mechanisms, TransferTraj effectively captures relative spatial correlations among trajectory points and identifies shared movement patterns in similar spatial contexts, thereby preventing bias toward specific regions. Second, we unify the input-output structure across different tasks to facilitate task transferability. Through a combination of randomly masking and recovering trajectory modalities or entire trajectory points, along with an effective pre-training mechanism, our model seamlessly transfers to various tasks without requiring retraining. For a discussion on the limitations and broader impacts of TransferTraj, please refer to Appendix A.

**Acknowledgment.** This work was supported by the Beijing Natural Science Foundation (Grant No. 4242029), A*STAR RIE2025 Manufacturing, Trade and Connectivity (MTC) Programmatic Fund (M24N6b0043), and ECNU Multifunctional Platform for Innovation (001).

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

Table 8: Statistics of datasets.

| Dataset | Chengdu | Xian | Porto |
|---|---|---|---|
| # Trajectories | 140,000 | 210,000 | 323, 481 |
| # POIs | 12,439 | 3,900 | 6,529 |
| # Road Segments | 4315 | 3392 | 9559 |
| # Sampling Intervals | $\sim$ 6s | $\sim$ 6s | 15s |
| Latitude range | $30.6552 \sim 30.7270$ | $34.2064 \sim 34.2802$ | $41.1405 \sim 41.1865$ |
| Longitude range | $104.0430 \sim 104.1265$ | $108.9172 \sim 109.0039$ | $-8.6887 \sim -8.5557$ |
| Size of training area (km$^2$) | 7.98 * 7.98 | 8.20 * 7.98 | 11.13 * 5.11 |

## A    Limitations and Broader Impacts

### A.1    Limitations

Our work faces challenges in cross-region classification tasks, such as trajectory-user linking or destination road segment prediction, due to the varying number of users and road networks across different regions. Future work will focus on developing a unified framework that enables the model to generalize more effectively across a broader range of tasks.

### A.2    Broader Impacts

We present a trajectory learning model that achieves both region and task transferability. This capability allows a single model to be applied across tasks without maintaining multiple model parameters, thereby significantly reducing storage and computation overhead. Moreover, the model can be directly transferred to other regions with limited data while maintaining strong performance, effectively alleviating overfitting issues caused by data scarcity.

## B    Experiment SetUp

### B.1    Dataset

Table 8 provides a statistical summary of Chengdu [5], Xian [6], and Porto[7], which exhibit variations in data scale, sampling frequency, spatial size, geographic origin, and the number of POIs and road segments. This heterogeneity renders them highly appropriate for a thorough assessment of the proposed model's regional transferability.

### B.2    Setting

For both datasets, we split the departure time of the trajectories chronologically into 8:1:1 ratios to create the training, validation, and testing sets. TransferTraj is implemented using the PyTorch framework and optimized with the Adam optimizer, with a learning rate set to 1e-3 and a batch size of 64. For the baseline models, we adopt the optimal hyperparameters as reported in their original papers. All baselines are trained for 50 epochs with an early stopping strategy based on a patience of 10 epochs. To retrieve the POIs and road segments surrounding a trajectory point, we set the distance thresholds $\varphi_{\text{dist}}^{\text{poi}} = 100$ meters and $\varphi_{\text{dist}}^{\text{road}} = 100$ meters. The three key hyperparameters of TransferTraj and their optimal values are $L = 2$, $d = 128$, and $C = 8$. A comprehensive analysis of the influence of these hyperparameters on model performance is provided in Appendix F. During pre-training, we randomly divide each trajectory into multiple non-overlapping segments, with the number of segments set to $0.2 \times |\mathcal{T}|$, where $|\mathcal{T}|$ represents the trajectory length. We then randomly select 40% of these segments and apply trajectory point masking to all points within the selected segments. For the remaining trajectory portions, we further select 20% of the trajectory points and apply either spatial or temporal modality masking. Experiments are conducted on Ubuntu 22.04

---

[5]`https://outreach.didichuxing.com/`
[6]`https://outreach.didichuxing.com/`
[7]`https://www.kaggle.com/competitions/pkdd-15-predict-taxi-service-trajectory-i`

servers equipped with Intel(R) Xeon(R) W-2155 CPUs and NVIDIA(R) TITAN RTX GPUs. Each experiment is repeated five times, and we report the mean and standard deviation of the evaluation metrics.

## B.3 Evaluation Metrics

For the TP and TR tasks, we evaluate the distance error between the predicted and ground-truth using Mean Absolute Error (MAE) and Root Mean Square Error (RMSE). For the OD TTE task, we use RMSE, MAE, and Mean Absolute Percentage Error (MAPE) to assess the discrepancy between the predicted and actual travel times. Lower values of these metrics indicate a better performance.

$$\text{RMSE} = \sqrt{\frac{1}{n}\sum_{i=1}^{n}(\hat{y}_i - y_i)^2}$$

$$\text{MAE} = \frac{1}{n}\sum_{i=1}^{n}|\hat{y}_i - y_i| \tag{6}$$

$$\text{MAPE} = \frac{1}{n}\sum_{i=1}^{n}\frac{|\hat{y}_i - y_i|}{y_i} \times 100\%$$

where $n$ denotes the total number of samples, $\hat{y}_i$ represents the predicted value, and $y_i$ is the target value.

## B.4 Baselines

For the **TP** task, we select the following seven trajectory representation baselines to compare model performance. We append two MLP layers after their output embedding to predict the destination location.

- **t2vec** [9]: Pre-trains the model by reconstructing original trajectories from low-sampling ones using a denoising auto-encoder.

- **Trembr** [11]: Constructs an RNN-based seq2seq model that recovers the road segments and time of the input trajectories.

- **CTLE** [23]: Pre-trains a bi-directional Transformer with two MLM tasks involving location and hour predictions. The trajectory representation is obtained by applying mean pooling on point embeddings.

- **Toast** [4]: Utilizes a context-aware node2vec model to generate segment representations and trains the model with an MLM-based task and a sequence discrimination task.

- **TrajCL** [1]: Introduces a dual-feature self-attention-based encoder and trains the model in a contrastive style using the InfoNCE loss.

- **START** [15]: Includes a time-aware trajectory encoder and a GAT that considers the transitions between road segments. The model is trained with both an MLM task and a contrastive task based on SimCLR loss.

- **LightPath** [40]: Constructs a sparse path encoder and trains it with a path reconstruction task and a cross-view and cross-network contrastive task.

We also compare the variant of the Trembr, START, and LightPath models without fine-tuning, where the pretrained encoder is frozen and only the MLP layers are fine-tuned. The variant is termed as Trembr (wo ft), START (wo ft), and LightPath (wo ft), respectively, as shown in the upper part of Table 1.

For the **TR** task, we select the following 7 trajectory recovery baselines. Including three relu-based methods and four learning-based methods.

- **Linear**: Assumes that the movement along the trajectory follows a uniform linear pattern. Missing trajectory points are imputed using linear interpolation between known points.

- **MPR** [6]: Divides the region into equally sized grids and estimates the most frequently traveled grid path between two known trajectory points. The center points of the inferred grids are then used to fill in the missing points.

- **TrImpute** [8]: Fills sparse trajectories using a crowd-wisdom-based algorithm, which leverages patterns learned from the behavior of multiple users.

- **DHTR** [33]: Employs a Seq2Seq framework to infer the grid sequences corresponding to missing trajectory points, followed by a Kalman filter to refine the recovered trajectory and enhance accuracy.

- **MTrajRec** [26]: Originally designed as a road network-constrained trajectory recovery model, which predicts road segment IDs and moving ratio. To enable region transferability, we modify its output layer to predict the latitude and longitude of missing points. The model adopts a GRU-based Seq2Seq architecture.

- **RNTrajRec** [5]: Similar to MTrajRec, we adapt the output to recover latitude and longitude coordinates instead of road segment IDs. It integrates trajectory sequences and road network information using a Transformer to capture spatio-temporal correlations for trajectory recovery.

- **MM-STGED** [36]: Utilizes graph-based methods to capture the semantic structure of trajectories and incorporates macro-level traffic conditions of regions to assist trajectory imputation.

For the **OD TTE** task, we select the following 8 origin-destination travel time estimation baselines.

- **RNE** [14]: Estimates distance between trajectory points by learning their latent embeddings, capturing spatial relations implicitly.

- **TEMP** [32]: Computes the average travel time of historical trajectories that are closely related in both spatial and temporal dimensions.

- **LR**: A simple linear regression model that maps input features to travel time based on temporal labels.

- **GBM**: A powerful non-linear regression model, implemented using XGBoost [2], capable of capturing complex feature interactions.

- **ST-NN** [16]: Simultaneously predicts travel distance and travel time for origin-destination pairs using a deep learning framework.

- **MURAT** [19]: Jointly predicts travel distance and time, while incorporating departure time as an auxiliary feature to enhance performance.

- **DeepOD** [43]: Leverages the correlation between input features and historical trajectories during training to improve prediction accuracy.

- **DOT** [24]: A two-stage framework that generates image-like representations of trajectories to estimate travel time through visual inference techniques.

### B.5 Variant of TransferTraj

The variations of the proposed model include:

- **wo TRIE**: Removes TRIE and uses vanilla transformer encoder to encode the sequence of trajectory points.

- **wo SC-MoE**: Removes the spatial context MoE.

- **wo POI modality**: Removes POI modality.

- **wo road network modality**: Removes road network modality.

## C The Proof of TRIE to Capture Relative Spatial Information

Given two trajectory points $p_i$ and $p_j$, associated with spatial modalities $(x_i, y_i)$ and $(x_j, y_j)$, respectively, we introduce a learnable spatiotemporal rotation matrix $\mathbf{R}_{\Phi(x,y)}$ to encode their relative spatial

information $(x_i - x_j, y_i - y_j)$ through matrix multiplication, where

$$\mathbf{R}_{\Phi(x,y)} = \begin{bmatrix} \cos\phi_1(x,y)\theta_1 & -\sin\phi_1(x,y)\theta_1 & \cdots & 0 & 0 \\ \sin\phi_1(x,y)\theta_1 & \cos\phi_1(x,y)\theta_1 & \cdots & 0 & 0 \\ \vdots & \vdots & \ddots & \vdots & \vdots \\ 0 & 0 & \cdots & \cos\phi_{d/2}(x,y)\theta_{d/2} & -\sin\phi_{d/2}(x,y)\theta_{d/2} \\ 0 & 0 & \cdots & \sin\phi_{d/2}(x,y)\theta_{d/2} & \cos\phi_{d/2}(x,y)\theta_{d/2} \end{bmatrix} \in \mathbb{R}^{d\times d} \tag{7}$$

Based on the matrix multiplication of the attention mechanism, we apply $\mathbf{R}_{\Phi(x,y)}$ to the query and key, allowing the model to effectively capture the relative information $\mathbf{R}_{\Phi(x_i-x_j,y_i-y_j)}$ between trajectory points. This design helps mitigate region-specific biases and improves the model's region transferability. The detailed derivation process is as follows:

$$
\begin{aligned}
\boldsymbol{q}_i \cdot \boldsymbol{k}_j^\top &= (\mathbf{R}_{\Phi(x_i,y_i)}\boldsymbol{W}_q\boldsymbol{e}_i)^\top \cdot (\mathbf{R}_{\Phi(x_j,y_j)}\boldsymbol{W}_k\boldsymbol{e}_j) \\
&= \boldsymbol{e}_i^\top \boldsymbol{W}_q^\top \mathbf{R}_{\Phi(x_i,y_i)}^\top (\mathbf{R}_{\Phi(x_j,y_j)}\boldsymbol{W}_k\boldsymbol{e}_j)
\end{aligned}
$$

$$
= \boldsymbol{e}_i^\top \boldsymbol{W}_q^\top \begin{bmatrix} \cos\phi_1(x_i,y_i)\theta_1 & \sin\phi_1(x_i,y_i)\theta_1 & \cdots & 0 & 0 \\ -\sin\phi_1(x_i,y_i)\theta_1 & \cos\phi_1(x_i,y_i)\theta_1 & \cdots & 0 & 0 \\ \vdots & \vdots & \ddots & \vdots & \vdots \\ 0 & 0 & \cdots & \cos\phi_{d/2}(x_i,y_i)\theta_{d/2} & \sin\phi_{d/2}(x_i,y_i)\theta_{d/2} \\ 0 & 0 & \cdots & -\sin\phi_{d/2}(x_i,y_i)\theta_{d/2} & \cos\phi_{d/2}(x_i,y_i)\theta_{d/2} \end{bmatrix}
$$

$$
\begin{bmatrix} \cos\phi_1(x_j,y_j)\theta_1 & -\sin\phi_1(x_j,y_j)\theta_1 & \cdots & 0 & 0 \\ \sin\phi_1(x_j,y_j)\theta_1 & \cos\phi_1(x_j,y_j)\theta_1 & \cdots & 0 & 0 \\ \vdots & \vdots & \ddots & \vdots & \vdots \\ 0 & 0 & \cdots & \cos\phi_{d/2}(x_j,y_j)\theta_{d/2} & -\sin\phi_{d/2}(x_j,y_j)\theta_{d/2} \\ 0 & 0 & \cdots & \sin\phi_{d/2}(x_j,y_j)\theta_{d/2} & \cos\phi_{d/2}(x_j,y_j)\theta_{d/2} \end{bmatrix} \boldsymbol{W}_k\boldsymbol{e}_j
$$

$$
= \boldsymbol{e}_i^\top \boldsymbol{W}_q^\top \begin{bmatrix} ① & ② & \cdots & 0 & 0 \\ ③ & ④ & \cdots & 0 & 0 \\ \vdots & \vdots & \ddots & \vdots & \vdots \\ 0 & 0 & \cdots & ⑤ & ⑥ \\ 0 & 0 & \cdots & ⑦ & ⑧ \end{bmatrix} \boldsymbol{W}_k\boldsymbol{e}_j
$$

$$
= \boldsymbol{e}_i^\top \boldsymbol{W}_q^\top \begin{bmatrix} \cos\phi_1[(x_i,y_i)-(x_j,y_j)]\theta_1 & \sin\phi_1[(x_i,y_i)-(x_j,y_j)]\theta_1 & \cdots & 0 & 0 \\ -\sin\phi_1[(x_i,y_i)-(x_j,y_j)]\theta_1 & \cos\phi_1[(x_i,y_i)-(x_j,y_j)]\theta_1 & \cdots & 0 & 0 \\ \vdots & \vdots & \ddots & \vdots & \vdots \\ 0 & 0 & \cos\phi_{d/2}[(x_i,y_i)-(x_j,y_j)]\theta_{d/2} & \sin\phi_{d/2}[(x_i,y_i)-(x_j,y_j)]\theta_{d/2} & \\ 0 & 0 & -\sin\phi_{d/2}[(x_i,y_i)-(x_j,y_j)]\theta_{d/2} & \cos\phi_{d/2}[(x_i,y_i)-(x_j,y_j)]\theta_{d/2} & \end{bmatrix} \boldsymbol{W}_k\boldsymbol{e}_j
$$

$$
= \boldsymbol{e}_i^\top \boldsymbol{W}_q^\top \mathbf{R}_{\Phi(x_i,y_i)-\Phi(x_j,y_j)}^\top \boldsymbol{W}_k\boldsymbol{e}_j
$$

$$
= \boldsymbol{e}_i^\top \boldsymbol{W}_q^\top \mathbf{R}_{\Phi(x_i-x_j,y_i-y_j)}^\top \boldsymbol{W}_k\boldsymbol{e}_j, \tag{8}
$$

where

$$
\begin{aligned}
① &= \cos\phi_1(x_i,y_i)\theta_1\cos\phi_1(x_j,y_j)\theta_1 + \sin\phi_1(x_i,y_i)\theta_1\sin\phi_1(x_j,y_j)\theta_1, \\
② &= -\cos\phi_1(x_i,y_i)\theta_1\sin\phi_1(x_j,y_j)\theta_1 + \sin\phi_1(x_i,y_i)\theta_1\cos\phi_1(x_j,y_j)\theta_1, \\
③ &= -\sin\phi_1(x_i,y_i)\theta_1\cos\phi_1(x_j,y_j)\theta_1 + \cos\phi_1(x_i,y_i)\theta_1\sin\phi_1(x_j,y_j)\theta_1, \\
④ &= \sin\phi_1(x_i,y_i)\theta_1\sin\phi_1(x_j,y_j)\theta_1 + \cos\phi_1(x_i,y_i)\theta_1\cos\phi_1(x_j,y_j)\theta_1, \\
⑤ &= \cos\phi_{d/2}(x_i,y_i)\theta_{d/2}\cos\phi_{d/2}(x_j,y_j)\theta_{d/2} + \sin\phi_{d/2}(x_i,y_i)\theta_{d/2}\sin\phi_{d/2}(x_j,y_j)\theta_{d/2}, \\
⑥ &= -\cos\phi_{d/2}(x_i,y_i)\theta_{d/2}\sin\phi_{d/2}(x_j,y_j)\theta_{d/2} + \sin\phi_{d/2}(x_i,y_i)\theta_{d/2}\cos\phi_{d/2}(x_j,y_j)\theta_{d/2}, \\
⑦ &= -\sin\phi_{d/2}(x_i,y_i)\theta_{d/2}\cos\phi_{d/2}(x_j,y_j)\theta_{d/2} + \cos\phi_{d/2}(x_i,y_i)\theta_{d/2}\sin\phi_{d/2}(x_j,y_j)\theta_{d/2}, \\
⑧ &= \sin\phi_{d/2}(x_i,y_i)\theta_{d/2}\sin\phi_{d/2}(x_j,y_j)\theta_{d/2} + \cos\phi_{d/2}(x_i,y_i)\theta_{d/2}\cos\phi_{d/2}(x_j,y_j)\theta_{d/2}
\end{aligned} \tag{9}
$$

## D   Performance Comparison in Task Transfer

Table 9 and 10 present the trajectory recovery performance on the Xi'an and Porto datasets. We observe that even using only the pretraining scheme (TransferTraj wo ft) in Section 4.2, our model outperforms the state-of-the-art trajectory recovery task baseline by 9.25% and 1.20%, demonstrating

Table 9: Overall performance of methods on trajectory recovery on the Xi'an dataset.

| Sampling Intervals | $\mu = \epsilon * 4$ | | $\mu = \epsilon * 8$ | | $\mu = \epsilon * 16$ | |
|---|---|---|---|---|---|---|
| Metric
Method | RMSE ↓
(meters) | MAE ↓
(meters) | RMSE ↓
(meters) | MAE ↓
(meters) | RMSE ↓
(meters) | MAE ↓
(meters) |
| Linear | 316.79 | 276.11 | 401.09 | 295.12 | 610.38 | 481.58 |
| MPR | 294.18 | 242.54 | 388.11 | 288.33 | 631.48 | 510.05 |
| TrImpute | 253.50 | 199.48 | 371.39 | 270.03 | 582.48 | 447.71 |
| DHTR | 286.19 ± 3.92 | 207.33 ± 3.23 | 394.18 ± 3.57 | 290.11 ± 4.54 | 612.38 ± 3.63 | 490.37 ± 4.02 |
| MTrajRec | 273.49 ± 6.63 | 195.19 ± 6.20 | 388.13 ± 7.38 | 279.19 ± 5.50 | 600.13 ± 7.91 | 464.67 ± 6.61 |
| RNTrajRec | 237.44 ± 3.89 | 168.58 ± 4.29 | 357.18 ± 4.59 | 266.22 ± 4.54 | 571.59 ± 3.15 | 438.48 ± 4.77 |
| MM-STGED | 214.58 ± 7.50 | 157.31 ± 9.80 | 331.10 ± 5.71 | 248.33 ± 8.14 | 544.14 ± 7.30 | 407.11 ± 5.25 |
| TransferTraj (wo pt) | 222.57 ± 3.42 | 171.93 ± 3.28 | 319.38 ± 3.25 | 238.47 ± 4.23 | 499.32 ± 2.87 | 381.48 ± 2.26 |
| TransferTraj (wo ft) | 201.48 ± 4.23 | 155.42 ± 4.73 | 290.19 ± 3.92 | 211.43 ± 4.19 | 471.70 ± 3.64 | 366.19 ± 4.23 |
| **TransferTraj** | 181.71 ± 3.91 | 132.83 ± 4.61 | 267.74 ± 4.55 | 196.01 ± 3.50 | 443.65 ± 3.92 | 322.54 ± 4.90 |

Red denotes the best result, and blue denotes the second-best result. ↓ means lower is better.

Table 10: Overall performance of methods on trajectory recovery on the Porto dataset.

| Sampling Intervals | $\mu = \epsilon * 4$ | | $\mu = \epsilon * 8$ | | $\mu = \epsilon * 16$ | |
|---|---|---|---|---|---|---|
| Metric
Method | RMSE ↓
(meters) | MAE ↓
(meters) | RMSE ↓
(meters) | MAE ↓
(meters) | RMSE ↓
(meters) | MAE ↓
(meters) |
| Linear | 430.59 | 329.48 | 523.69 | 415.77 | 762.48 | 561.39 |
| MPR | 404.19 | 310.04 | 504.11 | 396.28 | 722.48 | 537.28 |
| TrImpute | 372.95 | 288.28 | 486.11 | 377.41 | 693.58 | 511.49 |
| DHTR | 387.18 ± 2.02 | 293.19 ± 2.39 | 494.38 ± 3.79 | 384.59 ± 2.08 | 671.49 ± 3.37 | 492.33 ± 3.65 |
| MTrajRec | 394.19 ± 7.59 | 303.19 ± 8.44 | 488.41 ± 7.43 | 371.33 ± 4.97 | 682.51 ± 10.20 | 503.44 ± 7.07 |
| RNTrajRec | 366.19 ± 5.21 | 283.58 ± 4.61 | 474.19 ± 4.57 | 358.13 ± 4.15 | 649.18 ± 7.28 | 475.66 ± 5.96 |
| MM-STGED | 341.29 ± 8.15 | 252.44 ± 7.56 | 446.18 ± 11.27 | 328.34 ± 7.82 | 610.46 ± 13.30 | 448.11 ± 9.28 |
| TransferTraj (wo pt) | 361.49 ± 5.09 | 264.11 ± 4.66 | 485.28 ± 4.87 | 351.39 ± 4.33 | 622.48 ± 7.18 | 453.33 ± 4.89 |
| TransferTraj (wo ft) | 331.61 ± 4.52 | 258.39 ± 4.31 | 447.22 ± 5.82 | 321.39 ± 4.73 | 604.17 ± 4.38 | 431.00 ± 3.96 |
| **TransferTraj** | 316.71 ± 4.92 | 230.97 ± 3.83 | 431.66 ± 6.17 | 309.38 ± 5.24 | 578.71 ± 5.25 | 401.52 ± 3.24 |

Red denotes the best result, and blue denotes the second-best result. ↓ means lower is better.

strong task transferability. This capability is largely attributed to our modality and sub-trajectory masking and recovery strategies. Notably, in the most challenging trajectory recovery setting, where $\mu = 16 * \epsilon$, our model still surpasses the baseline by 11.68% and 2.42%. Furthermore, fine-tuning on the trajectory recovery task further boosts performance by an additional 9.54% and 5.56%, confirming the effectiveness of our model.

Table 11: Few-shot region transfer performance of methods on trajectory prediction.

| Dataset | Chengdu → Xian | | Chengdu → Porto | | Xian → Porto | | Xian → Chengdu | | Porto → Chengdu | | Porto → Xian | |
|---|---|---|---|---|---|---|---|---|---|---|---|---|
| Metric
Method | RMSE ↓
(meters) | MAE ↓
(meters) | RMSE ↓
(meters) | MAE ↓
(meters) | RMSE ↓
(meters) | MAE ↓
(meters) | RMSE ↓
(meters) | MAE ↓
(meters) | RMSE ↓
(meters) | MAE ↓
(meters) | RMSE ↓
(meters) | MAE ↓
(meters) |
| t2vec | 584.74 | 429.39 | 425.20 | 275.39 | 438.28 | 287.28 | 672.49 | 460.39 | 692.47 | 488.29 | 592.59 | 400.38 |
| Trembr | 583.69 | 459.49 | 387.29 | 237.49 | 395.20 | 236.20 | 573.59 | 443.68 | 592.58 | 460.29 | 571.38 | 439.10 |
| CTLE | 600.29 | 472.59 | 396.20 | 228.85 | 400.14 | 236.81 | 531.59 | 443.67 | 520.52 | 462.59 | 588.19 | 452.51 |
| Toast | 654.28 | 522.66 | 558.83 | 339.98 | 563.68 | 353.99 | 639.27 | 489.47 | 664.29 | 500.28 | 634.84 | 504.29 |
| TrajCL | 451.11 | 325.79 | 400.29 | 248.29 | 385.29 | 257.22 | 427.58 | 308.31 | 459.39 | 300.58 | 446.29 | 311.68 |
| START | 366.10 | 250.01 | 339.29 | 220.04 | 329.60 | 234.20 | 370.18 | 286.02 | 372.59 | 278.24 | 375.38 | 269.25 |
| LightPath | 648.30 | 392.68 | 442.48 | 286.80 | 473.59 | 299.00 | 651.60 | 452.60 | 682.59 | 473.59 | 652.55 | 384.18 |
| **TransferTraj** | 251.07 | 179.00 | 239.73 | 166.13 | 238.56 | 159.07 | 209.97 | 160.48 | 214.08 | 165.43 | 262.09 | 187.47 |

Red denotes the best result, and blue denotes the second-best result. ↓ means lower is better.

Table 12: Few-shot region transfer performance of methods on trajectory recovery on $\mu = \varepsilon * 4$.

| Dataset | Chengdu → Xian | | Chengdu → Porto | | Xian → Porto | | Xian → Chengdu | | Porto → Chengdu | | Porto → Xian | |
|---|---|---|---|---|---|---|---|---|---|---|---|---|
| Metric
Method | RMSE ↓
(meters) | MAE ↓
(meters) | RMSE ↓
(meters) | MAE ↓
(meters) | RMSE ↓
(meters) | MAE ↓
(meters) | RMSE ↓
(meters) | MAE ↓
(meters) | RMSE ↓
(meters) | MAE ↓
(meters) | RMSE ↓
(meters) | MAE ↓
(meters) |
| DHTR | 331.58 | 268.91 | 443.61 | 354.95 | 450.38 | 363.19 | 280.47 | 199.36 | 291.48 | 206.09 | 348.18 | 271.42 |
| MTrajRec | 326.13 | 254.44 | 431.44 | 341.49 | 428.47 | 337.18 | 266.39 | 185.10 | 261.96 | 177.39 | 335.55 | 264.19 |
| RNTrajRec | 275.26 | 213.52 | 414.68 | 301.38 | 405.18 | 311.47 | 251.26 | 177.29 | 268.19 | 171.54 | 271.38 | 201.33 |
| MM-STGED | 251.33 | 189.38 | 372.48 | 288.18 | 368.33 | 280.31 | 224.66 | 152.59 | 231.57 | 160.48 | 262.48 | 191.39 |
| **TransferTraj** | 205.18 | 156.27 | 342.57 | 260.31 | 352.48 | 264.29 | 163.68 | 120.58 | 173.29 | 122.41 | 210.48 | 162.52 |

Red denotes the best result, and blue denotes the second-best result. ↓ means lower is better.

Table 13: Few-shot region transfer performance of methods on trajectory recovery on $\mu = \varepsilon * 8$.

| Dataset | Chengdu → Xian | | Chengdu → Porto | | Xian → Porto | | Xian → Chengdu | | Porto → Chengdu | | Porto → Xian | |
|---|---|---|---|---|---|---|---|---|---|---|---|---|
| Metric | RMSE↓ | MAE↓ | RMSE↓ | MAE↓ | RMSE↓ | MAE↓ | RMSE↓ | MAE↓ | RMSE↓ | MAE↓ | RMSE↓ | MAE↓ |
| Method | (meters) | (meters) | (meters) | (meters) | (meters) | (meters) | (meters) | (meters) | (meters) | (meters) | (meters) | (meters) |
| DHTR | 491.70 | 337.06 | 574.81 | 429.25 | 593.25 | 440.85 | 388.72 | 273.19 | 387.55 | 273.65 | 498.81 | 346.79 |
| MTrajRec | 486.37 | 358.73 | 566.40 | 417.15 | 588.68 | 436.64 | 376.00 | 269.47 | 382.33 | 275.81 | 472.92 | 351.63 |
| RNTrajRec | 437.71 | 316.21 | 528.89 | 382.38 | 535.25 | 387.96 | 324.00 | 219.96 | 338.63 | 237.19 | 446.29 | 329.03 |
| MM-STGED | 383.08 | 285.06 | 491.11 | 369.49 | 503.86 | 375.76 | 279.22 | 208.54 | 275.35 | 204.97 | 374.99 | 291.16 |
| **TransferTraj** | 294.90 | 230.39 | 461.66 | 327.96 | 473.73 | 339.08 | 203.04 | 158.42 | 198.43 | 152.80 | 288.77 | 215.32 |

Red denotes the best result, and blue denotes the second-best result. ↓ means lower is better.

Table 14: Few-shot region transfer performance of methods on trajectory recovery on $\mu = \varepsilon * 16$.

| Dataset | Chengdu → Xian | | Chengdu → Porto | | Xian → Porto | | Xian → Chengdu | | Porto → Chengdu | | Porto → Xian | |
|---|---|---|---|---|---|---|---|---|---|---|---|---|
| Metric | RMSE↓ | MAE↓ | RMSE↓ | MAE↓ | RMSE↓ | MAE↓ | RMSE↓ | MAE↓ | RMSE↓ | MAE↓ | RMSE↓ | MAE↓ |
| Method | (meters) | (meters) | (meters) | (meters) | (meters) | (meters) | (meters) | (meters) | (meters) | (meters) | (meters) | (meters) |
| DHTR | 673.85 | 539.70 | 752.11 | 569.15 | 763.71 | 574.97 | 548.24 | 442.99 | 553.62 | 450.61 | 682.19 | 549.39 |
| MTrajRec | 684.97 | 546.07 | 765.80 | 579.63 | 759.57 | 568.07 | 537.96 | 428.73 | 535.42 | 431.40 | 692.10 | 555.81 |
| RNTrajRec | 638.11 | 519.24 | 711.28 | 538.19 | 724.88 | 550.82 | 492.89 | 362.15 | 482.34 | 365.84 | 646.12 | 538.46 |
| MM-STGED | 584.21 | 446.28 | 674.94 | 497.29 | 683.06 | 508.39 | 433.77 | 328.19 | 446.36 | 325.39 | 591.61 | 452.40 |
| **TransferTraj** | 472.74 | 363.38 | 611.86 | 449.68 | 605.98 | 453.94 | 318.14 | 235.84 | 327.70 | 244.86 | 466.49 | 358.62 |

Red denotes the best result, and blue denotes the second-best result. ↓ means lower is better.

Table 15: Few-shot region transfer performance of methods on OD TTE.

| Dataset | Chengdu → Xian | | | Chengdu → Porto | | | Xian → Porto | | |
|---|---|---|---|---|---|---|---|---|---|
| Metric | RMSE↓ | MAE↓ | MAPE↓ | RMSE↓ | MAE↓ | MAPE↓ | RMSE↓ | MAE↓ | MAPE↓ |
| Method | (minutes) | (minutes) | (%) | (minutes) | (minutes) | (%) | (minutes) | (minutes) | (%) |
| LR | 7.105 | 6.353 | 19.396 | 4.993 | 2.990 | 34.689 | 4.612 | 2.867 | 34.195 |
| GBM | 6.445 | 5.529 | 17.693 | 4.461 | 2.748 | 29.573 | 4.628 | 2.770 | 30.197 |
| ST-NN | 5.914 | 5.011 | 16.003 | 4.067 | 2.660 | 26.899 | 4.333 | 2.776 | 27.250 |
| MuRAT | 5.610 | 4.509 | 14.536 | 3.819 | 2.443 | 25.686 | 3.813 | 2.511 | 21.063 |
| DeepOD | 4.973 | 3.935 | 12.996 | 3.603 | 2.390 | 20.371 | 3.639 | 2.414 | 19.617 |
| DOT | 4.771 | 3.372 | 11.096 | 3.087 | 2.003 | 17.693 | 2.915 | 2.218 | 17.502 |
| **TransferTraj** | 4.286 | 2.928 | 9.376 | 2.565 | 1.533 | 15.378 | 2.447 | 1.598 | 15.935 |
| Dataset | Xian → Chengdu | | | Porto → Chengdu | | | Porto → Xian | | |
| Metric | RMSE↓ | MAE↓ | MAPE↓ | RMSE↓ | MAE↓ | MAPE↓ | RMSE↓ | MAE↓ | MAPE↓ |
| Method | (minutes) | (minutes) | (%) | (minutes) | (minutes) | (%) | (minutes) | (minutes) | (%) |
| LR | 4.731 | 3.815 | 20.886 | 4.875 | 3.883 | 19.941 | 6.703 | 5.983 | 19.091 |
| GBM | 4.537 | 3.561 | 18.386 | 4.525 | 3.619 | 18.004 | 6.538 | 5.429 | 17.021 |
| ST-NN | 4.301 | 3.417 | 16.480 | 4.296 | 3.372 | 15.257 | 5.637 | 4.811 | 16.029 |
| MuRAT | 4.228 | 3.322 | 15.498 | 4.324 | 3.297 | 14.397 | 5.306 | 4.602 | 14.667 |
| DeepOD | 3.998 | 3.175 | 13.319 | 4.012 | 3.179 | 13.744 | 4.992 | 3.998 | 14.074 |
| DOT | 3.827 | 2.991 | 11.967 | 3.709 | 3.204 | 12.481 | 4.536 | 3.029 | 12.636 |
| **TransferTraj** | 3.532 | 2.730 | 9.990 | 3.691 | 2.811 | 9.803 | 4.339 | 2.874 | 10.946 |

Red denotes the best result, and blue denotes the second-best result. ↓ means lower is better.

# E  Performance Comparison in Few-shot Region Transfer

We conduct a regional transferability experiment under the assumption that a small amount of data is available in the target region. To evaluate model performance in this setting, we first train the models on a source region and then fine-tune them using 5,000 trajectories from the target region. The results across three downstream tasks and all datasets are shown in Table 11, 12, 13, 14 and 15. Our model consistently achieves the best performance, with improvements of 33.68%, 18.08%, and 13.07% on TP, TR, and OD TTE tasks, respectively. Compared to the zero-shot transfer setting discussed in Section 5.1, baseline models show significant performance gains after fine-tuning, as they focus on learning region-specific representations. In contrast, the superior transferability of our model stems from its carefully designed region-agnostic feature representation, which enables robust adaptation across different geographic areas.

# F  Hyperparameter Study

We analyze three key hyperparameters: the hidden state dimension $d$ of the model, the number of block $L$ of stacked TRIE and SC-MoE, the value of $k$ and the number of experts $C$ in the SC-

Table 16: Hyperparameter range and optimal value.

| Parameter | Range |
|---|---|
| The number of hidden state $d$ | 32, 64, 128, 256, 512 |
| The number of layer $L$ of stacked TRIE and SC-MoE | 1, 2, 3, 4, 5, 6 |
| The number of experts in the SC-MoE | $c_1 : k = 1, C = 6; c_2 : k = 2, C = 6; c_3 : k = 4, C = 6;$ $c_3 : k = 1, C = 8; c_4 : k = 2, C = 8; \underline{c_5} : k = 4, C = 8; c_6 : k = 6, C = 8$ |
| The distance threshold of POI and road segment $\varphi_{\text{dist}}^{\text{poi}}$ and $\varphi_{\text{dist}}^{\text{road}}$ | 50, 100, 150, 200 |

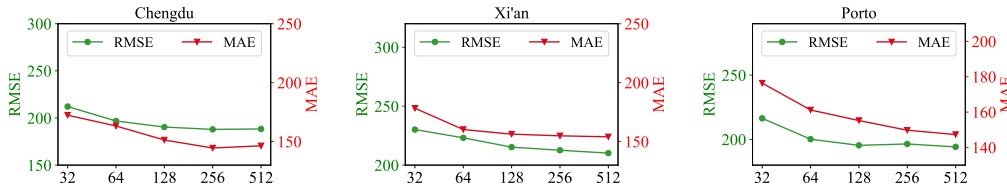

Figure 3: Hyperparameter analysis of the $d$.

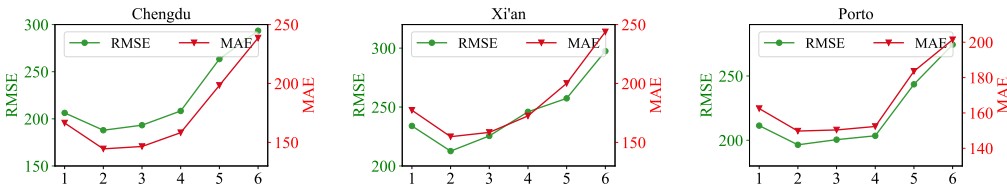

Figure 4: Hyperparameter analysis of the $L$.

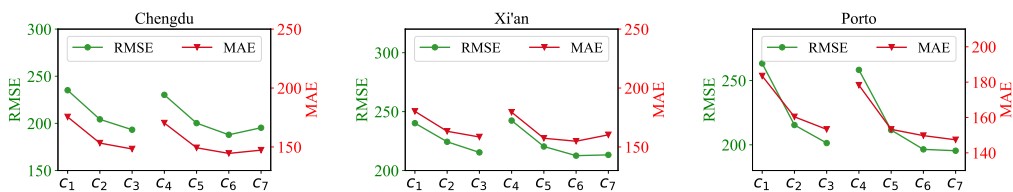

Figure 5: Hyperparameter analysis of the value of $k$ and the number of expert $C$.

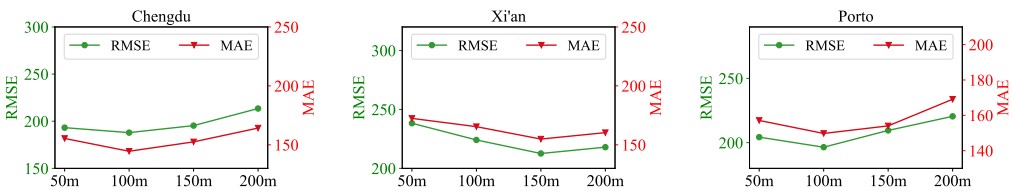

Figure 6: Hyperparameter analysis of the distance threshold of POI and road segment $\varphi_{\text{dist}}^{\text{poi}}$ and $\varphi_{\text{dist}}^{\text{road}}$.

MoE, and the distance threshold of POI and road segment $\varphi_{\text{dist}}^{\text{poi}}$ and $\varphi_{\text{dist}}^{\text{road}}$. The values of these hyperparameters, along with their optimal value, are summarized in the Table 16. Figure 3, 4, 5, and 6 present the experimental results of these hyperparameters on three datasets for trajectory prediction tasks. The results show that varying $L$ significantly impacts model performance, and $L$ has an optimal value of 2; beyond that, the model becomes overly complex and harder to train. The optimal value

for $d$ is 256; smaller values limit the model's learning capability, while larger values offer minimal performance gains and decrease computational efficiency.

For the SC-MoE, setting $k = 4$ and the number of experts $C = 8$ yields optimal performance. This can be attributed to the fact that too few experts limit the model's ability to capture trajectory movement patterns, as they can handle only limited spatial context. On the other hand, increasing the number of experts excessively raises model complexity, complicating the training process and ultimately compromising performance.

For the $\varphi_{\text{dist}}^{\text{poi}}$ and $\varphi_{\text{dist}}^{\text{road}}$, Chengdu and Porto datasets achieve optimal performance at 100 meters, while Xi'an performs best at 150 meters. This difference likely stems from Xi'an's sparser POI and road network distribution ($\sim 111.44$ POIs and road segments per km$^2$ in Xi'an vs. $\sim 263.10$ in Chengdu and $\sim 282.87$ in Porto), where a smaller radius does not capture sufficient spatial context. Performance first improves then declines with increasing distance, indicating that insufficient POIs/road segments provide inadequate contextual information while excessive numbers introduce redundancy.

