# OpenReview forum: "TransferTraj: A Vehicle Trajectory Learning Model for Region and Task Transferability"
_NeurIPS.cc/2025/Conference — NeurIPS 2025 oral_

### Official Review · Reviewer_rKX5 · 2025-06-26

**Clarity:** 3
**Significance:** 3
**Originality:** 3
**Rating:** 5
**Confidence:** 4

**Summary:**

TransferTraj proposes a unified trajectory-learning framework that simultaneously enables region-level and task-level transfer. Without retraining, the mask-and-recover pre-training objective can be applied to three generative downstream tasks—trajectory prediction, trajectory recovery, and OD travel-time estimation. Experiments on Chengdu, Xi’an, and Porto taxi datasets show that TransferTraj outperforms strong baselines, confirming both its effectiveness and scalability.

**Questions:**

See weakness.

**Ethical Concerns:**

["NO or VERY MINOR ethics concerns only"]

**Limitations:**

Yes

**Paper Formatting Concerns:**

None.

**Quality:**

3

**Strengths And Weaknesses:**

Strengths

(1) The authors tackle both region and task transferability in a single framework, reducing the need to train and maintain multiple region- or task-specific models.

(2) Extensive experiments provide strong empirical evidence for the ability of TransferTraj.

(3) A unified masking-and-recovery pre-training scheme enables zero-shot usage on three heterogeneous generative tasks without re-training prediction heads.



Weaknesses

(1) The evaluation of TP task only focuses on the precision of the trajectories’ destinations and ignores the intermediate trajectory, which weakens the claim of accurate trajectory prediction.

(2) The unified input and output scheme of three tasks is pivotal to the task transferability, which is just stated in Appendix. Without an explicit, worked example in the main body, it will hurt clarity and reproducibility.

---

> ### Author Rebuttal · Authors · 2025-07-30
>
> Thank you for your insightful feedback! We are grateful for your acknowledgment of our work. Below, we address each of your concerns and questions in detail:
>
>
>
> > [W1] The evaluation of TP task only focuses on the precision of the trajectories’ destinations and ignores the intermediate trajectory, which weakens the claim of accurate trajectory prediction.
>
> We conducted additional experiments evaluating trajectory prediction performance for predicting 1, 2, 3, 4, and 5 future trajectory points on the Chengdu dataset. The experiment results show that the prediction error increases with the number of predicted steps. However, our model consistently outperforms baselines across all prediction lengths, demonstrating its effectiveness in trajectory prediction tasks.
>
> |                           |      | START  | LightPath | TransferTraj (Ours) |
> | ------------------------- | ---- | ------ | --------- | ------------------- |
> | Further 1-step prediction | RMSE | 58.59  | 63.37     | **22.83**           |
> |                           | MAE  | 36.35  | 54.52     | **15.94**           |
> | Further 2-step prediction | RMSE | 142.55 | 177.03    | **63.86**           |
> |                           | MAE  | 93.64  | 168.40    | **43.62**           |
> | Further 3-step prediction | RMSE | 218.21 | 368.26    | **117.50**          |
> |                           | MAE  | 163.44 | 201.18    | **98.18**           |
> | Further 4-step prediction | RMSE | 296.94 | 427.34    | **162.47**          |
> |                           | MAE  | 207.06 | 284.87    | **120.10**          |
> | Further 5-step prediction | RMSE | 333.10 | 553.27    | **187.91**          |
> |                           | MAE  | 240.40 | 360.86    | **144.53**          |
>
>
>
> > [W2] The unified input and output scheme of three tasks is pivotal to the task transferability, which is just stated in Appendix. Without an explicit, worked example in the main body, it will hurt clarity and reproducibility.
>
> Due to page limitations, the unified input-output scheme details for three tasks are provided in Appendix C. If the paper is accepted, we will move this critical content to the main text for better clarity and reproducibility.

---

> > ### Comment · Reviewer_rKX5 · 2025-08-04
> >
> > Thank you for the clarifications and added results. Your response addresses my concerns, so I maintain my positive score and recommendation for acceptance.

---

> > > ### Author Response · Authors · 2025-08-04
> > >
> > > Thank you for taking the time to review our rebuttals and for your positive score. We are pleased that our responses have effectively addressed your questions.

---

### Official Review · Reviewer_dG9Z · 2025-06-26

**Clarity:** 3
**Significance:** 3
**Originality:** 3
**Rating:** 4
**Confidence:** 4

**Summary:**

This article proposes TransferTraj, a deep learning-based architecture designed to facilitate transfer across regions and tasks in the context of trajectory prediction. The model addresses domain discrepancies that arise between different geographic regions and related tasks, such as prediction, recovery, and travel time estimation. To achieve this, it introduces the Region-Transferable Trajectory Encoder (RTTE), which encodes spatial and temporal information from trajectories. RTTE includes two key components: TRIE, which captures relative spatial relationships between trajectory points, and SC-MoE, a mechanism that detects movement patterns based on local context by leveraging similarities in urban or road environments. Additionally, the article proposes a task-transfer scheme based on masking and recovering trajectory modalities or points. The proposed approach is evaluated on three real-world datasets and compared against multiple baseline methods, showing improvements in accuracy and generalization capability.

**Questions:**

In the proposed method, there are some important implementation details that remain unclear. For instance, in the definition of the POI modality: ” The POI modality $Pi = {l | dis(l, pi) ≤ \underset{\text{dist}}{\varphi^{\text{poi}}}}$ is defined as the set of all POIs lying within a distance $\underset{\text{dist}}{\varphi ^{\text{poi}}}$ of the trajectory point $pi$.”
What specific distance metric is used for dis(l, pi)? Is it the Haversine distance, Euclidean, or another geodesic measure appropriate for geographic coordinates?

In the same context, $\underset{\text{dist}}{\varphi ^{\text{poi}}}$ and $\underset{\text{dist}}{\varphi ^{\text{road}}}$ are introduced as distance thresholds for selecting POIs and road segments around each trajectory point. However, it is not explained how these hyperparameters were selected. Were these values fixed across all datasets, or tuned for each region individually? What was the rationale or methodology behind the selection of these thresholds?

Regarding the Transformer-based multimodal architecture, it is unclear whether the model was trained entirely from scratch or built upon a pre-existing encoder. Could the authors clarify whether a standard Transformer (e.g., RoFormer, ViT, etc.) was adapted, or whether the encoder was implemented from the ground up? Additionally, how many layers and attention heads are used in the RTTE module?

Although Figure 1 provides a schematic overview of the proposed architecture, its connection to the textual description is limited. I suggest aligning the narrative more explicitly with the labeled components of the figure as the model is described. In particular, the integration of the four modalities (spatial, temporal, POI, road network) is conceptually important but currently somewhat unclear. A more detailed mapping between the components in the figure and their function in the pipeline would greatly improve clarity.

**Ethical Concerns:**

["NO or VERY MINOR ethics concerns only"]

**Final Justification:**

I will raise my score to an acceptance. The authors have satisfactorily addressed my questions, and I have read the comments from other reviewers, which indicate that the paper has the potential to be accepted

**Limitations:**

The paper presents certain technical limitations, particularly regarding the lack of detailed experimentation and insufficient information on the training and comparison of baseline models. For further discussion, please refer to the “Weaknesses” and “Questions” sections.

**Paper Formatting Concerns:**

No formatting concerns.

**Quality:**

3

**Strengths And Weaknesses:**

Strengths:
- The article addresses a real and relevant problem: the generalization of trajectory models to new regions and tasks.
- It proposes an architecture that integrates key components to improve transferability across regions and tasks, combining several established deep learning techniques such as mixture-of-experts, attention mechanisms, and masking + recovery strategies. Additionally, the use of a rotation matrix to capture spatiotemporal information is a novel and valuable idea.
- Multiple baseline models are considered for comparison, which supports the empirical evaluation of the proposed method.

Weaknesses:
- Although several baselines are used for comparison, the paper does not specify how they were trained or tuned. It is not clear whether pre-trained weights were reused or whether hyperparameters were fairly adjusted, which raises concerns about the fairness of the comparisons.
- The paper lacks implementation details such as the number of layers, embedding size, batch size, and task-specific learning rates. These aspects are only superficially mentioned in the appendices and not explained in the main text.

---

> ### Author Rebuttal · Authors · 2025-07-30
>
> Thank you for your insightful feedback! We are grateful for your acknowledgment of our idea. Below, we address each of your concerns and questions in detail:
>
>
>
> > [W1] Although several baselines are used for comparison, the paper does not specify how they were trained or tuned. It is not clear whether pre-trained weights were reused or whether hyperparameters were fairly adjusted, which raises concerns about the fairness of the comparisons.
>
> For baseline settings, we adopted hyperparameters reported in the original papers and retrained models from scratch on our datasets. For trajectory representation learning baselines, we first pre-trained their trajectory encoders from scratch, then fine-tuned both encoders and output heads following standard trajectory representation learning protocols. For baselines used in trajectory recovery and origin-destination travel time estimation tasks, we conducted end-to-end training from scratch. Baseline settings are described in Appendix B.2 (lines 448-450), and we will include these clarifications in the revised version.
>
>
>
> > [W2] The paper lacks implementation details such as the number of layers, embedding size, batch size, and task-specific learning rates.
>
> Due to page limitations, implementation details are provided in Appendix B.2. Our settings are: number of layers $L = 2$, embedding size $d = 128$, batch size 64, and learning rate $1 \times 10^{-4}$. These choices are justified through hyperparameter experiments presented in Appendix G (Figures 3-5). If accepted, we will move this content to the main text.
>
>
>
> > [Q1] What specific distance metric is used for dis(l, pi)? Is it the Haversine distance, Euclidean, or another geodesic measure appropriate for geographic coordinates?
>
> We use Haversine distance to calculate the distance between the trajectory point and the POI in meters.
>
>
>
> > [Q2] How to select the hyperparameter of  $\varphi_{\text{dist}}^{\text{poi}}$ and $\varphi_{\text{dist}}^{\text{road}}$ ?
>
> The parameters $\varphi_{\text{dist}}^{\text{poi}}$ and $\varphi_{\text{dist}}^{\text{road}}$ are region-specific and adjustable. We added a hyperparameter experiment on Chengdu, Xi'an, and Porto dataset in trajectory prediction task.
>
> |                                                              | Chengdu    |            | Xi'an      |            | Porto      |            |
> | ------------------------------------------------------------ | :--------- | ---------- | ---------- | ---------- | ---------- | ---------- |
> | $\varphi_{\text{dist}}^{\text{poi}}, \varphi_{\text{dist}}^{\text{road}}$ | RMSE       | MAE        | RMSE       | MAE        | RMSE       | MAE        |
> | 50m                                                          | 193.18     | 155.38     | 238.29     | 172.41     | 204.29     | 157.11     |
> | 100m                                                         | **187.91** | **144.53** | 224.13     | 165.39     | **196.46** | **149.75** |
> | 150m                                                         | 195.36     | 152.49     | **212.62** | **154.86** | 209.51     | 154.03     |
> | 200m                                                         | 213.48     | 164.20     | 217.98     | 160.32     | 220.47     | 169.10     |
>
> The experiment results show that Chengdu and Porto datasets achieve optimal performance at 100 meters, while Xi'an performs best at 150 meters. This difference likely stems from Xi'an's sparser POI and road network distribution (~111.44 POIs and road segments per $\text{km}^2$ in Xi'an vs. ~263.10 in Chengdu and ~282.87 in Porto), where a smaller radius does not capture sufficient spatial context. Performance first improves then declines with increasing distance, indicating that insufficient POIs/road segments provide inadequate contextual information while excessive numbers introduce redundancy. This hyperparameter analysis will be included in the revised version.
>
>
>
> > [Q3] Regarding the Transformer-based multimodal architecture, it is unclear whether the model was trained entirely from scratch or built upon a pre-existing encoder. Could the authors clarify whether a standard Transformer (e.g., RoFormer, ViT, etc.) was adapted, or whether the encoder was implemented from the ground up? Additionally, how many layers and attention heads are used in the RTTE module?
>
> The model is trained entirely from scratch, adapted from RoFormer by replacing the original rotary position encoding with our learnable spatiotemporal position encoding and substituting the fully connected layer with SC-MoE. These modifications enable better capture of relative relationships between trajectory points and adaptive extraction of movement patterns based on surrounding spatial context, enhancing regional transferability.
>
> According to our hyperparameter experiments (Appendix G, Figure 4), the RTTE model uses 2 layers with 1 attention head.
>
>
>
> > [Q4] Although Figure 1 provides a schematic overview of the proposed architecture, its connection to the textual description is limited.
>
> We appreciate your careful observation and constructive suggestions. We will revise Figure 1 to better align the visual components with the textual description, providing clearer mapping between the four modalities (spatial, temporal, POI, road network) and their integration in the pipeline. Specifically, we will present a more intuitive pipeline that illustrates the association between trajectory points and the four modalities, as well as the encoding and mixing processes of these modalities. For the TRIE module, we will clearly show its capability to capture the relative relationships between trajectory points. Additionally, we will update the figure’s labels to ensure consistency with the textual description.

---

> > ### Comment · Reviewer_dG9Z · 2025-08-03
> >
> > Thank you for addressing all of my questions. I have carefully reviewed your responses, as well as those provided to the other reviewers. I now have no further questions and feel prepared to make my decision.
> > Best regards,

---

> ### Author Response · Authors · 2025-08-04
>
> Thank you for taking the time to consider the other reviews and rebuttals. We are glad that we were able to answer your questions.

---

### Official Review · Reviewer_sVcu · 2025-06-29

**Clarity:** 3
**Significance:** 3
**Originality:** 4
**Rating:** 5
**Confidence:** 5

**Summary:**

This paper presents an innovative vehicle trajectory learning framework that supports both region and task transfer. It addresses the limitations of existing embedding-based methods, which usually require retraining when applied to new regions or tasks. The model uses four modalities and two key modules TRIE and SC-MoE to capture relative spatial relationships and adapt to different movement patterns across regions. The authors also propose a unified input-output scheme that reformulates various tasks as a masking and recovery problem, making it possible to pretrain the model in a task-agnostic way. Experiments show that TransferTraj performs exceptionally well on three real-world datasets.

**Questions:**

1.Compared to traditional trajectory representation learning models, what are the core advantages of TransferTraj in terms of task transferability?

2.Why can the POI and road network modalities generalize across different regions, given that their spatial distributions and semantics may differ?

3. In the task-transferable input-output scheme, how are the masking rates for modalities and trajectory points determined?

**Ethical Concerns:**

["NO or VERY MINOR ethics concerns only"]

**Final Justification:**

The article is particularly good. rebuttal perfectly solved my problem. I suggest accepting this article.

**Limitations:**

Yes

**Quality:**

4

**Strengths And Weaknesses:**

Strengths:
1.The paper presents an interesting topic and introduces a novel unified framework that supports both region and task transfer within a single model, without requiring retraining. The proposed RTTE module and the task-transferable input-output scheme are both innovative and effective. Together, they enable the model to generalize well to other regions, different spatial contexts, and various trajectory-related tasks.

2.The authors provide a theoretical derivation to show that the proposed module is effective in capturing the relative spatial relationships between trajectory points.

3.The paper is well-structured and logically organized. It clearly explains the motivation and challenges of learning relative spatial information, capturing dynamic movement patterns, and designing a unified input-output format.

4.The author conducts comprehensive experiments on zero-shot, few-shot, and task transfer scenarios across multiple tasks and datasets, proving that the proposed models can outperform the SOTA. The code and data are publicly available, which ensures reproducibility.

Overall, I think this paper offers an interesting and novel solution for transferable vehicle trajectory learning, but a few weaknesses can be highlighted:

Weakness:

1.Theoretically, trajectory representation learning models can also achieve task transferability by attaching task-specific heads. What are the key advantages of TransferTraj compared to these methods?

Since POIs and road network structures vary significantly across regions, the authors should further clarify how these two modalities are encoded.

---

> ### Author Rebuttal · Authors · 2025-07-30
>
> Thank you for your positive and insightful comments. We are delighted that the reviewer found our motivation interesting and reasonable. Below, we address each of your concerns and questions in detail:
>
>
>
> > [W1 & Q1] The key advantages of TransferTraj compared to traditional trajectory representation learning models.
>
> Trajectory representation learning models pre-train an encoder to obtain universal trajectory embeddings, requiring separate training of task-specific output heads, which incurs additional training costs. Furthermore, our experiments demonstrate that trajectory representation learning requires joint fine-tuning of both the pre-trained encoder and output head when applied to various tasks (as shown in Table 1).
>
> In contrast, TransferTraj eliminates the need for task-specific output heads and additional training, directly adapting to various tasks through our well-designed pre-training scheme while achieving excellent experimental results. Additionally, our model transfers directly to other regions without retraining while maintaining strong performance, whereas existing trajectory representation learning models require separate training for each region.
>
>
>
> > [W2 & Q2] Further clarify how POI and road network modalities are encoded, and explain why they can be generalized across different regions.
>
>
>
> When encoding POIs and road networks, we utilize textual descriptions rather than ID embeddings to capture semantic information. As described in the Preliminaries section, we represent POIs using their name, type, and address as text descriptions, while road segments are described using their name, type, and length. These text descriptions are input into OpenAI's pre-trained text embedding model to generate fixed-dimension vectors. Since the extracted semantic information depends solely on POI and road network attributes and is independent of specific regions, this approach guarantees transferability across different geographical areas.
>
>
>
> > [Q3] In the task-transferable input-output scheme, how are the masking rates for modalities and trajectory points determined?
>
> During pre-training, we randomly divide each trajectory into multiple non-overlapping segments, with the number of segments set to $0.2 \times T$, where $T$ represents the trajectory length. We then randomly select 40% of these segments and apply trajectory point maskinxg to all points within the selected segments. For the remaining trajectory portions, we further select 20% of the trajectory points and apply either spatial or temporal modality masking. These parameter settings will be clarified in the revised version.

---

> > ### Comment · Reviewer_sVcu · 2025-08-03
> >
> > The authors address my previous concern. I will keep my score as "Accept".

---

> ### Author Response · Authors · 2025-08-03
>
> Thank you for your response. Your suggestion is valuable and we will follow your guidance to incorporate the feedback in the revised version.

---

### Official Review · Reviewer_qUMg · 2025-07-01

**Clarity:** 3
**Significance:** 3
**Originality:** 3
**Rating:** 5
**Confidence:** 4

**Summary:**

This paper proposes TransferTraj, a novel vehicle trajectory learning model designed for both transferability across regions and downstream tasks. The author introduces RTTE and spatial, temporal, POI, and road network modalities to address region transfer by modeling relative spatial relationships using a rotary position matrix and dynamically routing trajectories to experts based on local spatial context. Additionally, the paper introduces the unified input and output structures for various tasks via modality and points masking and recovery, enabling task transfer after pre-training. Extensive experiments on three real-world datasets show SOTA performance in task transfer and zero-shot region transfer.

**Questions:**

1.The experiments are conducted with k=4 and C=8, but Figure 2 shows that the number of activated experts in different-density regions mostly concentrates around 2, 3, and 4. Does this suggest that some experts may be redundant?

2.When the sampling rate is set to $\mu=16\epsilon$, both the proposed model and the baseline experience a significant increase in error. What might be the underlying cause of this phenomenon?

3.In the pre-training section, how are the lengths of the masked start and end segments defined? how is the modality mask of trajectory points determined?

The TRIE module is designed to address generalization with varying trajectory lengths. What is the distribution of trajectory lengths across different regions? Is the model sensitive to trajectory length or sampling frequency?

**Ethical Concerns:**

["NO or VERY MINOR ethics concerns only"]

**Final Justification:**

All my concerns have addressed and I keep a postive score on this paper.

**Limitations:**

yes

**Paper Formatting Concerns:**

No concerns

**Quality:**

3

**Strengths And Weaknesses:**

Strengths:
1.The paper introduces a novel problem formulation and is the first to incorporate both regional and task transferability into a single model, laying the foundation for the development of a trajectory base model.

2.This paper presents a well-designed and technically sound architecture. The TRIE and SC-MoE modules effectively handle spatial feature differences and regional context distributions. A unified pre-training strategy enables multi-task learning without introducing additional parameters.

3.Experiments are comprehensive and rigorous, covering three tasks, three datasets, and diverse transfer scenarios. Ablation and efficiency studies further validate the model’s effectiveness and lightweight design.

4.The paper is well-structured and clearly articulates its research objectives, methodology, and conclusions.

Weakness:
1.Although the authors claim that the TRIE module addresses the challenge of varying trajectory lengths across regions, they do not provide a clear explanation of how this is achieved. Additionally, the paper lacks an analysis of trajectory length distribution differences across datasets to support this claim.

2.The author terms the top-K gating mechanism "prevent the movement patterns from being represented by the same set of experts", however, it does not explain why or how this mechanism successfully enforces expert diversity.

3.There are some inconsistencies in the writing. For example, the term "trajectory modality" is used interchangeably as "modality" and "feature" throughout the paper. The masked spatial and temporal modalities ( p^{ms} and p^{mt} in Section 4.2) are not clearly illustrated in the figures.

---

> ### Author Rebuttal · Authors · 2025-07-30
>
> Thank you for your positive and insightful comments. We are delighted that the reviewer found our motivation interesting and reasonable. Below, we address each of your concerns and questions in detail:
>
>
>
> > [W1 & Q4] Explaining why TRIE can address the challenge of varying trajectory lengths and the analysis of trajectory length distribution.
>
> **About TRIE's generalization to varying trajectory lengths:** TRIE achieves length generalization through its learnable spatial-temporal rotation matrix encoding method. Each trajectory point is encoded as a rotation vector on the $d$-dimensional plane, where relative spatial distance information between trajectory points is naturally embedded through cosine and sine functions. This design ensures that angular differences between positions remain both periodic and continuous, maintaining robustness when processing sequences longer than those seen during training. Therefore, TRIE can seamlessly extend to longer sequences during inference while preserving accurate relative position representation and computational stability.
>
> **About trajectory length distribution:** The datasets exhibit different trajectory length distributions: Chengdu has lengths ranging from 20-120 points with an average distance of 4.057km; Xi'an ranges from 20-120 points with an average distance of 5.253km; Porto ranges from 20-100 points with an average distance of 8.284km. These statistical differences in length distributions make cross-dataset transfer challenging. Our zero-shot transfer experiments demonstrate that TRIE maintains strong generalization ability across these varying distributions, validating its effectiveness.
>
>
>
> > [W2] The author terms the top-K gating mechanism "prevent the movement patterns from being represented by the same set of experts", however, it does not explain why or how this mechanism successfully enforces expert diversity.
>
> Introducing the top-K gating mechanism in the routing stage can break the model's over-reliance on specific experts, increase the randomness of routing decisions, and improve the overall utilization of experts. In standard MoE architectures, the router tends to repeatedly select a few high-performing experts, causing different movement patterns to be assigned to the same subset of experts. This leads to expert overload and overfitting, weakening the model's expressiveness and generalization ability. The top-K gating mechanism prevents this by forcing the model to explore more diverse expert combinations during training, ensuring better utilization of all available experts. This design enables the model to capture fine-grained differences in movement patterns more effectively. Each expert specializes in a specific subspace of movement patterns, enhancing the model's ability to distinguish and generalize across complex and diverse trajectory behaviors.
>
>
>
> > [W3] There are some inconsistencies in the writing.
>
> Thank you for your observations. We will revise them in the revision.
>
>
>
> > [Q1] The experiments are conducted with k=4 and C=8, but Figure 2 shows that the number of activated experts in different-density regions mostly concentrates around 2, 3, and 4. Does this suggest that some experts may be redundant?
>
> Although experts 2, 3, and 4 are frequently activated, each operates within different spatial density contexts. Expert 3 is primarily activated in high-density areas, expert 2 in medium-density areas, and expert 4 in low-density areas. This pattern demonstrates that the model effectively allocates different spatial contexts to different experts through the routing mechanism. Furthermore, the combination of activated experts varies significantly across density levels, confirming that our model adaptively activates appropriate experts based on spatial context, enhancing its modeling capability. Additionally, our hyperparameter study in Figure 5 examines the effect of the number of experts C and activations K on model performance. The experimental results show that reducing the number of experts leads to noticeable performance degradation. Therefore, the current configuration of K=4 and C=8 represents a balanced and optimal choice.
>
>
>
> > [Q2] When the sampling rate is set to $\mu=16\epsilon$, both the proposed model and the baseline experience a significant increase in error. What might be the underlying cause of this phenomenon?
>
> This phenomenon occurs because the missing rate of trajectory points is extremely high. When $\mu = 16 \times \epsilon$, 93.75% of trajectory points are missing, resulting in severely limited available information. Consequently, both our model and baselines experience degraded performance compared to other sampling rates. However, our model still maintains significant improvement over baselines even under these challenging conditions, further demonstrating its effectiveness.
>
>
>
> > [Q3] In the pre-training section, how are the lengths of the masked start and end segments defined? how is the modality mask of trajectory points determined?
>
> During pre-training, we randomly divide each trajectory into multiple non-overlapping segments, with the number of segments set to $0.2 \times T$, where $T$ represents the trajectory length. We then randomly select 40% of these segments and apply trajectory point masking to all points within the selected segments. For the remaining trajectory portions, we further select 20% of the trajectory points and apply either spatial or temporal modality masking. These parameter settings will be clarified in the revised version.

---

> > ### Comment · Reviewer_qUMg · 2025-08-01
> >
> > The authors address my previous concern. I will keep my score as "Accept".

---

> ### Author Response · Authors · 2025-08-01
>
> Thank you for your response. Your suggestion is valuable and we will follow your guidance to incorporate the feedback in the revised version.

---

### Note · Authors · 2025-08-12

We sincerely thank the AC and all reviewers for their time and valuable feedback. **We are grateful that our novelty, motivation, problem setting, and writing have been acknowledged, with all reviewers giving positive scores.** We appreciate the opportunity to present and further clarify our work.

## Summary of Contributions
We propose a vehicle trajectory learning model supporting both regional and task transferability. Once pre-trained on one region, the model can be directly applied to other regions and diverse trajectory tasks, enabling efficient zero-shot and few-shot adaptation. This transferability not only improves computational efficiency but also maintains strong performance with limited training data.

Technically, we introduce a trajectory point initialization method to eliminate dependence on specific regions, employ learnable spatio-temporal rotation matrix to capture relative spatial information while removing regional bias, and design an SC-MoE to adaptively select movement experts based on spatial context. In addition, we propose a unified input–output paradigm with a pre-training strategy, enabling seamless adaptation to multiple tasks.

## Broader Impact
By addressing both regional and task transferability, our work offers new perspectives for trajectory learning, reducing storage and computation costs, sustaining high performance with limited data, and alleviating overfitting risks from data scarcity.

## Key Points in Rebuttal
1. We demonstrate that our model achieves superior efficiency in training cost and performance compared to existing trajectory representation learning models (Reviewer sVcu).
2. We justify the setting of POI and road network distance (Reviewer dG9Z), explaining how spatial context influences movement patterns and how expert activation distribution (Reviewer dG9Z, qUMg). We also clarify how TRIE generalizes to trajectories of varying lengths (Reviewer qUMg).
3. We clarify specific settings (Reviewer qUMg, sVcu, dG9Z) and show that as prediction steps increase, both our model and baselines have more errors, with our model consistently outperforming baselines (Reviewer rKX5).

## Future Revision Commitment
We will further refine the writing and clarify technical details in the final version.

## Closing Acknowledgement
Once again, we sincerely thank the AC and all reviewers for their constructive feedback and support, which have greatly improved the clarity and rigor of our work.

---

### Decision · Program_Chairs · 2025-09-17

**Decision:**

Accept (oral)

**Comment:**

This paper proposes TransferTraj, a novel vehicle trajectory learning model designed for both transferability across regions and downstream tasks. It is the first to incorporate both regional and task transferability into a single model, laying the foundation for the development of a trajectory base model. Further, the novelty of the proposed method is recognized by all reviewers. Extensive experiments are conducted to validate the proposed effectiveness of the proposed method. The authors' rebuttals have fully  addressed all reviewers' concerns and all reviewers recommended the acceptance of the paper.